# Nucleus accumbens D1- and D2-expressing neurons control the balance between feeding and activity-mediated energy expenditure

Roman Walle [1] ✉, Anna Petitbon[1], Giulia R. Fois [2], Christophe Varin [3], Enrica Montalban[1], Lola Hardt[1], Andrea Contini[1], Maria Florencia Angelo[1], Mylène Potier[1,4], Rodrigue Ortole[1], Asma Oummadi[1], Véronique De Smedt-Peyrusse[1], Roger A. Adan[5,6], Bruno Giros [7,8], Francis Chaouloff[9,10], Guillaume Ferreira [1], Alban de Kerchove d'Exaerde[3], Fabien Ducrocq[1], François Georges [2] & Pierre Trifilieff [1] ✉

Accumulating evidence points to dysregulations of the Nucleus Accumbens (NAc) in eating disorders (ED), however its precise contribution to ED symptomatic dimensions remains unclear. Using chemogenetic manipulations in male mice, we found that activity of dopamine D1 receptor-expressing neurons of the NAc core subregion facilitated effort for a food reward as well as voluntary exercise, but decreased food intake, while D2-expressing neurons have opposite effects. These effects are congruent with D2-neurons being more active than D1-neurons during feeding while it is the opposite during running. Chronic manipulations of each subpopulations had limited effects on energy balance. However, repeated activation of D1-neurons combined with inhibition of D2-neurons biased behavior toward activity-related energy expenditure, whilst the opposite manipulations favored energy intake. Strikingly, concomitant activation of D1-neurons and inhibition of D2-neurons precipitated weight loss in anorexia models. These results suggest that dysregulations of NAc dopaminoceptive neurons might be at the core of EDs.

Overeating/obesity-related pathologies and anorexia nervosa (AN) are multifactorial diseases resulting from an interaction between genetic and environmental factors. These eating disorders (ED) are characterized by psychiatric symptoms coupled to metabolic alterations, making them increasingly viewed as "metabo-psychiatric" diseases[1,2]. Genome-wide association studies (GWAS) reveal an inverse genetic correlation between AN and obesity[3], as well as overlapping environmental factors[4]. Dimensional approaches through the Research Domain Criteria (RDoC) framework to EDs reveals alterations in common domains such as negative (stress, anxiety) and positive (motivation, reward processing) valence systems[5]. In particular, beyond the opposite feeding patterns, which might be related to inverse alterations of food-related reward valuation[5], reward processing tends to be dysregulated in opposite manners in overeating and AN. In particular, an increased drive to exercise due to enhanced reinforcing value of physical activity is a typical feature of AN that persists after remission and might even precede the development of the disease[6-8]. On the other hand, obesity is associated with physical

[1]Université de Bordeaux, INRAE, Bordeaux INP, NutriNeuro, 33000 Bordeaux, France. [2]Univ. Bordeaux, CNRS, IMN, UMR5293 F-33000, Bordeaux, France. [3]Laboratory of Neurophysiology, ULB Neuroscience Institute, WELBIO, Université Libre de Bruxelles (ULB), Brussels, Belgium. [4]Bordeaux Sciences Agro, F-, 33175 Gradignan, France. [5]Department of Translational Neuroscience, UMC Utrecht Brain Center, University Medical Center Utrecht, Universiteitsweg 100, 3584CG Utrecht, Netherlands. [6]Altrecht Eating Disorders Rintveld, Zeist, the Netherlands. [7]Department of Psychiatry, Douglas Hospital, McGill University, Montreal, QC, Canada. [8]Université de Paris Cité, INCC UMR 8002, CNRS; F-75006, Paris, France. [9]Endocannabinoids and NeuroAdaptation, NeuroCentre INSERM U1215, 33077 Bordeaux, France. [10]Université de Bordeaux, 33077 Bordeaux, France. ✉e-mail: roman.walle@live.fr; pierre.trifilieff@inrae.fr

hypoactivity even after recovery[9]. These differential alterations of physical activity in EDs have been proposed to be linked to the rewarding impact of exercise and/or the motivation to engage in physical activity[10,11]. Such overlapping symptomatic dimensions in EDs suggest opposite dysfunctions within common brain networks, especially the reward system.

In regard to its well-established role in reward processing, motivation and locomotor activity, the nucleus accumbens (NAc) has received particular attention as it stands as a hub within the reward system, computing sensory, cognitive and reward information to modulate goal-directed behaviors[12]. Functional imaging reveals that NAc activity is regulated in opposite manner in AN and obese subjects in response to food cues[13]. Hence, this suggests that the reward system is hyperresponsive in AN while hyporesponsive in obese subjects[14], at least in relation to food-related stimuli. In the same vein, studies aiming at addressing the integrity of dopamine transmission found decreased and increased striatal dopaminergic D2/3 receptor binding in obese subjects[15,16] and in AN individuals[17], respectively, even though these latter imaging data in obesity and AN are controversial[18–20].

Altogether, these findings point to dopamine-dependent transmission of the NAc playing a key role in the pathophysiology of EDs. In the NAc, most neurons are accounted for by dopaminoceptive medium spiny neurons (MSNs) which are equally divided in two subpopulations expressing either D1 (D1-MSNs) or D2 (D2-MSNs) receptors. The respective roles of D1- and D2-MSNs of the NAc have been the subject of extensive investigation for their implication in food-related reward processing and motivational processes[21–27]. Recent studies also highlight a direct role of both MSN subpopulations of the NAc in the regulation of food intake. In particular, D1-MSNs of the NAc shell have been shown to negatively regulate feeding[28–31], although chronic blockade of their synaptic outputs prevented the development of obesity[32]. On the other hand, striatal overexpression of the D2 receptor restricted to the developmental period facilitates weight gain due to decreased energy expenditure[33], while downregulation of striatal D2 receptor at adulthood leads to compulsion towards palatable food[34]. Conversely, its overexpression in the NAc induces rapid weight loss in female mice exposed to the so-called Activity-Based Anorexia (ABA) model[35]. Interestingly, such effects of manipulations of D2 receptor expression mainly result from physical activity-related energy expenditure rather than from a direct effect on food intake[35–37]. However, how MSN subpopulations act in concert to regulate the balance between energy intake and activity-mediated expenditure and to which extent dysregulation of MSN network could be sufficient to mimic specific obesity- or AN-related dimensions remain unknown.

To solve this issue, we used imaging to characterize the signature of activity of D1- and D2-neurons during feeding and voluntary running. We assessed the impact of acute or chronic manipulation of the activity of D1- and D2-neurons on incentive processes as well as the balance between feeding and voluntary activity-mediated energy expenditure using a chemogenetic approach. The rational for using such long-lasting manipulation was to mimic to some extent the changes in resting state functional connectivity of the NAc described in obesity[38,39] and AN[40,41]. We provide evidence that the balance between D1- and D2-neurons of the NAc differentially determines the decision to engage in food seeking, feeding or voluntary exercise. Our data point to the NAc being at the crossroad between reward processing and energy balance, further supporting that dysfunction of that network could be a main player in ED etiology.

## Results

### D1-neurons and D2-neurons of the NAc respectively facilitate and diminish incentive processes towards palatable food

Alterations in food reward valuation are main features of EDs[5]. Because the respective roles of NAc D1- and D2-MSNs in food-oriented incentive processes was recently reconsidered[23–27], we aimed at assessing the consequence of long-lasting manipulation of either D1- or D2-expressing neurons – as a proxy for pathological dysregulation of these neuronal subpopulations - in motivation to exert effort to obtain a food reward. Activating or inhibitory DREADDs were expressed in either D1- or D2-expressing neurons of the NAc core (see supplementary information) through stereotaxic injections in D1- or D2-cre mice (Fig. 1A). Of note, as previously described[22,42], we found very little viral-mediated expression of cre-dependent DREADD, if any, in ChAT-positive neurons (SF1A), supporting that cholinergic interneurons were mainly spared by DREADD manipulations. In accordance with the literature, D2-neurons strongly project to the ventral pallidum (VP), while D1-neurons of the NAc core project to both the VP and the SNc/VTA (Fig. 1B, G).

We found that chemogenetic inhibition or activation of D2-neurons increased and decreased performance in a progressive ratio task, respectively (Fig. 1C–E; SF1B). There was no effect on licking patterns during consumption of a palatable sweetened milk solution (SF1C) - a proxy for hedonic reactivity -, nor on motoric abilities (Fig. 2F, SF1D). However, locomotor activity was increased and decreased by D2-neurons inhibition and activation, respectively (SF1E). Regarding D1-neurons, both chemogenetic inhibition and activation decreased performance in the progressive ratio task (SF1F-G). This was accompanied by an inability to increase press rate as a function of ratio requirement in the PRx2 task for the DREADD Gq group, together with increased locomotion, but no change in hedonic reactivity (SF1G, I, J, K). Decreasing the dose of CNO recapitulated the effects in PRx2, though sparing the ability to increased press rate as a function of ratio requirement (Fig. 1H–K). Use of the DREADD ligand JHU37160 (J60) led to the same findings, confirming that this was not related to a non-specific CNO effect (SF2H).

Considering the previously described satietogenic effect of NAc D1-neurons activation, we hypothesized that the diminution of lever pressing could be related to a decreased hunger drive rather than blunted motivational performance per se.

### NAc D1-neurons and D2-neurons exert opposite effects on food consumption

We therefore tested the effect of chemogenetic manipulations of NAc neuronal subpopulations on food intake. In the absence of effort required – i.e. when the reward was delivered in a pavlovian conditioning setting (Fig. 2A) –, D2-neurons inhibition in food-deprived animals decreased consumption of the sweetened milk reward, while their activation had the opposite effect (Fig. 2B and SF2A). The mirroring pattern was observed under manipulations of D1-neurons (Fig. 2B and SF2A). To confirm the effect on food intake per se, we assessed the effects of modulating NAc subpopulations activity on *ad libitum* food consumption (SF2B). Inhibition of D2-neurons acutely decreased chow consumption, hence resulting in lower cumulative food intake over 48 hours (SF2B). D2-neurons activation had the opposite effect, though the increased chow consumption was delayed (SF2B). We interpret this latter finding as the result of an initial overshadowing of the orexigenic effect of D2-neurons activation by the blunted motivational drive, decreasing the willingness to exert effort to retrieve the food pellets despite the increased hunger drive. This is supported by the observation that in the case of easily-accessible liquid food (milk), under D2-neurons activation, food consumption immediately increases as animals mostly stay at the dipper to consume (Fig. 2B and SF2A). Such effects on food intake resulted in a significant weight loss for D2-neurons inhibition and weight gain for D2-neurons activation, these being maintained for 48 hours (SF2B). Of note, these effects were not related to differences in peripheral glucose tolerance (SF2D). Manipulation of D1-neurons activity resulted in opposite findings with their inhibition leading to acute increased food intake whilst their activation strongly decreased it (SF2B), with no effect on

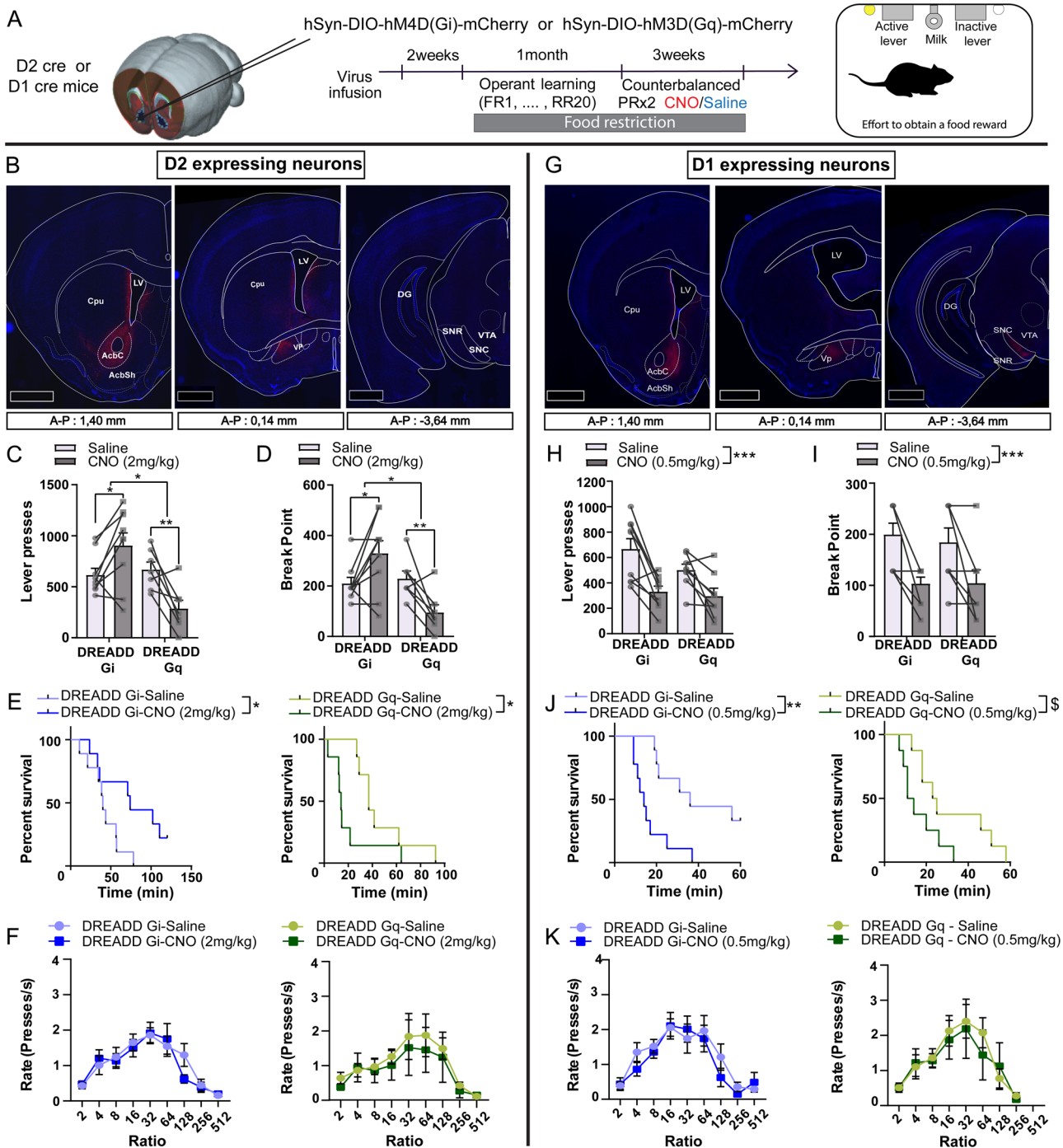

**Fig. 1 | Effect of chemogenetic manipulations of D2- and D1-neurons of the NAc on the motivational dimension of feeding behavior.** **A** Experimental timeline. **B** Representative coronal sections for mCherry expression after injection of a DREADD-mCherry-carrying viral vector in the NAc core of D2-cre mice (left), highlighting projections in the ventral pallidum (VP, middle) but not the midbrain (right). Scale bars 1 mm. **C**−**F** Effects of chemogenetic-mediated D2-neurons activation (DREADD Gq, $n = 7$) and inhibition (DREADD Gi, $n = 9$) under administration of CNO at 2 mg/kg on the number of reinforced lever presses (**C**; Two-way RM ANOVA; Interaction DREADD x injection $F_{(1, 14)} = 19.29$, $p = 0.0006$; Post-hoc saline vs CNO: Gi, $p = 0.0258$; Gq, $p = 0.0096$), breakpoint (**D**; Two-way RM ANOVA; Interaction DREADD x injection $F_{(1, 14)} = 19.92$, $p = 0.0017$; Post-hoc saline vs CNO: Gi, $p = 0.0310$; Gq, $p = 0.0327$) the percentage of mice that keep on responding as a function of time (**E**; Logrank test; Gi $p = 0.0337$; Gq $p = 0.0118$) and the ratio requirement plotted as the rate of lever presses as a function of PR ratios (**F**). **G** Representative coronal sections for mCherry expression after injection of DREADD-mCherry-carrying viral vector in the NAc core of D1-cre mice (left),

highlighting projections in the ventral pallidum (VP, middle) and the midbrain (right). **H**−**K** Effects of chemogenetic-mediated D1-neurons activation (DREADD Gq, $n = 8$) and inhibition (DREADD Gi, $n = 9$) under administration of CNO at 0.5 mg/kg on the number of reinforced lever presses (**H**; Two-way RM ANOVA; CNO effect $F_{(1, 15)} = 28.22$ $p = 0.0001$), breakpoint (**I**; Two-way RM ANOVA; CNO effect $F_{(1, 15)} = 25.29$ $p = 0.0001$) the percentage of mice that keep on responding as a function of time (**J**; Logrank test; Gi $p = 0.0028$; Gq $p = 0.0560$) and the ratio requirement plotted as the rate of lever presses as a function of PR ratios (**K**). Data are averages of 2 PR sessions under CNO or saline administrations in a within-subject, counterbalanced manner. AcbC Accumbens core; AcbSh Accumbens shell; Cpu Caudate Putamen; DG Dentate Gyrus; LV Lateral ventricle; SNC *Substantia nigra pars compacta*; SNR *Substantia nigra pars reticulata*; Vp Ventral Pallidum; VTA Ventral tegmental area. $: 0.05 < p < 0.1$; *: $p < 0.05$; **: $p < 0.01$; ***: $p < 0.001$; Error bars = s.e.m. Detailed statistics are displayed in Supplementary Table 2. Source data are provided as a Source Data file.

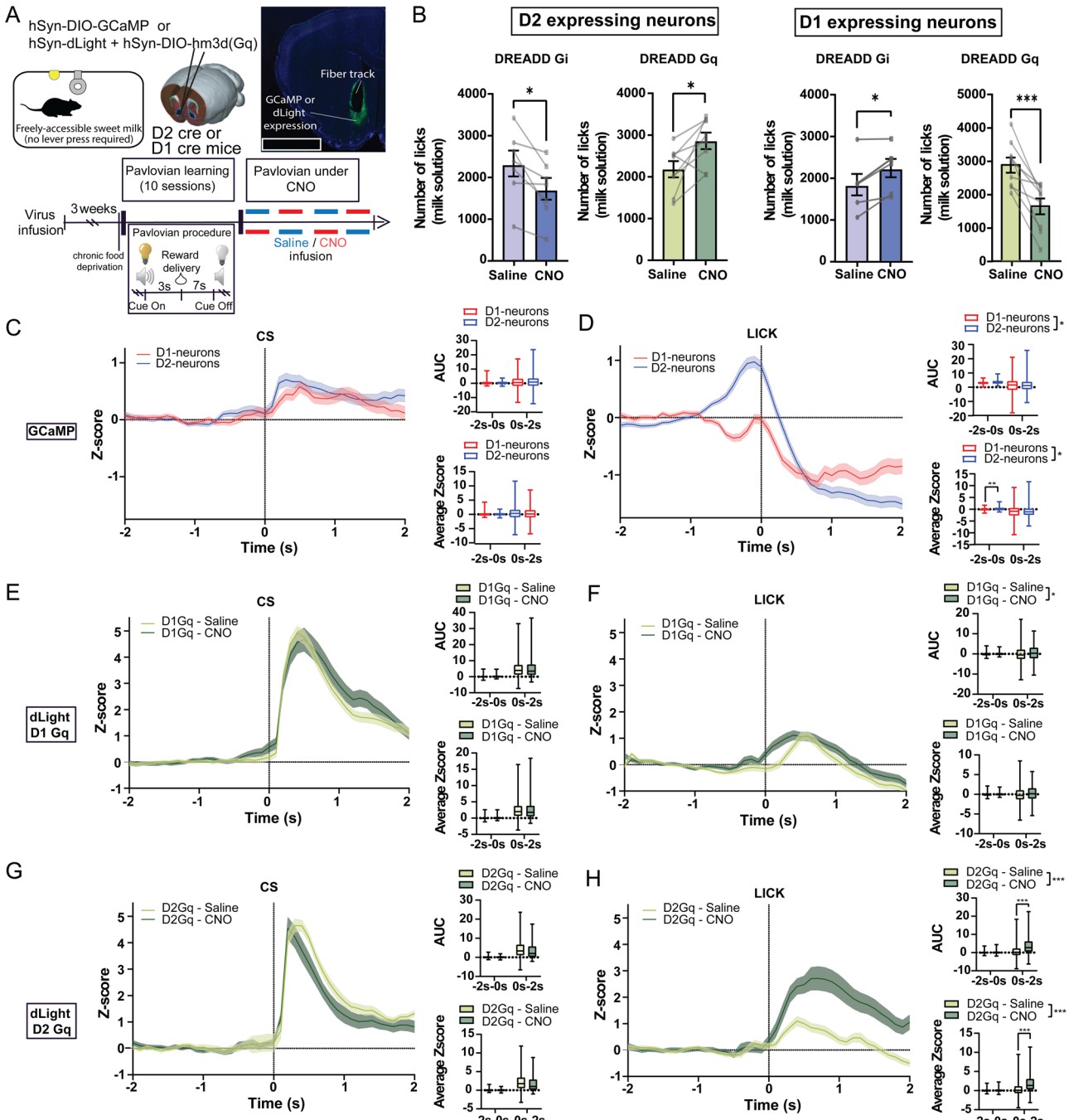

**Fig. 2 | D1- and D2-neurons of the NAc are differentially activated during reward consumption and oppositely regulate reward consumption in a pavlovian conditioning paradigm. A** Schematic representation of the procedure and representative image of sensors expression and fiber implantation. Bar scale 2.5 mm. **B** Effect of chemogenetic inhibition (DREADD Gi, D2-neurons $n = 7$, D1-neurons $n = 6$) and activation (DREADD Gq, $n = 8$, D1-neurons $n = 9$) of either D2-neurons or D1-neurons on the number of licks on the trough that delivers the reward (Two-sided paired t-test; D2 Gi $p = 0.0179$, D2 Gq $p = 0.0281$, D1 Gi $p = 0.0127$, D1 Gq $p = 0.0008$). **C, D** Comparison of bulk calcium dynamics in D1- ($n = 5$ mice) and D2-neurons ($n = 5$ mice) aligned to the onset of the conditioned stimuli (CS, D1 events=528; D2 events=591) (**C**), or to the onset of feeding (lick, D1 events=593; D2 events=660) and quantification of the AUC and average z-score (**D**), measured through expression of the calcium sensor GcAMP in the NAc, in a pavlovian conditioning paradigm (Two-way ANOVA; Lick AUC: Group effect $F_{(1, 2502)} = 9.477$, $p = 0.0021$; Lick Average Zscore: Group effect $F_{(1, 2502)} = 4.301$,

$p = 0.0382$; Post-hoc D1 vs D2 2s-0s $p = 0.0043$). **E, F** Effect of chemogenetic activation of D1-neurons (D1 Gq, $n = 4$ mice) on dopamine dynamics and quantification of the AUC and average z-score, measured through the dopamine sensor dLight coupled with fiber photometry in the NAc, aligned with CS onset (E; events Saline = 252; events CNO = 168) or lick (F; events Saline = 271; events CNO = 184) (Two-way RM ANOVA; AUC CNO effect $F_{(1, 455)} = 4.010$, $p = 0.0458$). **G, H** Effect of chemogenetic activation of D2-neurons (D2 Gq, $n = 3$ mice) on calcium dynamics measured through the dopamine sensor dLight coupled with fiber photometry in the NAc, aligned with CS onset (**G**; events Saline = 170; events CNO = 105) or lick onset (**H**; events Saline = 192; events CNO = 112) (Two-way RM ANOVA; AUC CNO effect $F_{(1, 302)} = 31.40$, $p = 0.0001$; Post-hoc Sal vs CNO 0-2 s $p = 0.0001$; Average Z score CNO effect $F_{(1, 302)} = 32.06$ $p = 0.0001$; Post-hoc Sal vs CNO 0-2 s $p = 0.0001$). $p < 0.05$; **: $p < 0.01$; ***: $p < 0.001$; Error bars = s.e.m. Detailed statistics are displayed in Supplementary Table 2. Source data are provided as a Source Data file.

glucose metabolism (SF2D). However, these acute effects on food intake were not accompanied by significant changes in weight (SF2B).

As food intake and willingness to work for a food reward were regulated in opposite manners by D1- and D2-neuron manipulations, we next aimed at determining whether an increase in the motivational valence of food might lead to different outcomes on food intake. We therefore challenged mice in a fasting/refeeding protocol wherein animals were allowed access to palatable food (high-fat) after an overnight fasting (SF2C). We found effects comparable to those measured during *ad libitum* chow intake, i.e. D2-neurons inhibition decreased food intake while their activation had a delayed orexigenic effect (SF2C). Manipulation of D1-neurons triggered the opposite effects (SF2C). Both high (2 mg/kg) and low (0.5 mg/kg) doses of CNO resulted in similar outcomes, with milder effects at 0.5 mg/kg. We therefore kept CNO at the dose of 2 mg/kg for the following experiments.

These opposite effects of D1 and D2-neurons manipulations on food intake made us hypothesize that each neuronal subpopulation might display opposite pattern of activity during food consumption. Using GCaMP-mediated calcium imaging coupled with fiber photometry in a pavlovian conditioning paradigm (Fig. 2A), we found that, in trained animals, while the relative bulk activity of D1- and D2-neurons was comparable in response to the reward-predicting cues (Fig. 2C), D2-neurons were significantly more active than D1-neurons – that even tended to be inhibited – when animals engaged in reward consumption (Fig. 2D). Together with our chemogenetic experiments, these findings support that the balance towards higher activity of D2-neurons over D1-neurons promotes feeding. However, reward prediction seems to be comparably encoded by these two neuronal subpopulations, suggesting that alterations in feeding as a result of chemogenetic manipulations of D1- and D2-neurons do not originate from a perturbation of associative processes between the reward and predicting cues. To further explore this hypothesis, we assessed the effect of chemogenetic activation of D1- or D2-neurons on mesoaccumbens dopamine dynamics given the role of this pathway in associative learning. We first confirmed through electrophysiological recordings that activation of D1- and D2-neurons exert opposite effect on the firing of dopamine cells in the VTA, namely inhibition and activation, respectively (SF2E). We next assessed the impact on mesoaccumbens dopamine dynamics in a pavlovian conditioning paradigm, through expression of the dopamine sensor dLight1.3 coupled with fiber photometry (Fig. 2A). In trained animals, activation of either D1- or D2-neurons had no effect on dopamine response to food cues (Fig. 2E, G). During reward consumption, activation of D1-neurons did not affect dopamine dynamics (Fig. 2F), while activation of D2-neurons enhanced dopamine response (Fig. 2H). Altogether, these latter findings further support that chemogenetic manipulations of NAc neuronal subpopulations directly impact feeding response rather than associative processes and that these effects could be partially mediated by an influence on dopamine transmission.

## Opposite balance between NAc D1- and D2-neurons activities during food consumption and voluntary physical activity

As mentioned above, alterations in voluntary physical activity are typical symptoms of EDs that go in opposite directions with food consumption. We hypothesized that running and feeding would be accompanied by an opposite balance in the activity of D1- and D2-neurons. We therefore measured the relative bulk activity pattern of D1- and D2-neurons of the NAc during food consumption and wheel running through recording of calcium dynamics coupled with fiber photometry (Fig. 3A, B, SF3A-C). In food- or wheel running-deprived animals, bulk activity of both D1- and D2-neuron were comparably increased when animals engaged towards either food consumption or the running wheel (Fig. 3A, B, SF3A-B). Strikingly, the activity of both

D1- and D2-neurons dropped during food reward consumption, with the relative activity of D2-neurons remaining significantly higher than that of D1-neurons (Fig. 3A, SF3A–B). In opposite, during running, the relative activity of D1-neurons remained significantly higher than that of D2-neurons (Fig. 3B, SF3A–B). Interestingly, when animals stopped running, D2-neurons activity remained higher than that of D1-neurons (SF3C). Altogether, these findings support that the balance of activity between D1- and D2-neurons of the NAc is biased towards D2-neurons during food consumption while voluntary wheel running correlates with higher activity of D1-neurons over D2-neurons.

## The effects of acute chemogenetic NAc neuron manipulations on food intake occur at the expense of voluntary physical activity

Based on the aforementioned findings we then hypothesized that altering the balance between D1- and D2-neurons could bias choice towards eating or running. We performed a choice protocol in which fasted mice had free access to food and a running wheel (Fig. 3C, SF3D-G). The acute satietogenic effect of D2-neurons inhibition was accompanied by increased wheel running distance, while activation of D2-neurons resulted in a strong decrease in physical activity with no significant effect on food intake (Fig. 3D, SF3D-E), in accordance with the delayed effect of such manipulation on feeding (SF2B-C). In opposite, inhibition of D1-neurons increased and decreased food intake and wheel running, respectively, while the satietogenic effect of D1-neurons activation was accompanied by increased wheel running (Fig. 3E; SF3F-G).

## Chronic activation of NAc D2-neurons leads to increased fat mass related to decreased voluntary physical activity

Altogether, our findings suggest that the opposite effects of D1- and D2-neurons manipulation on food intake do not reflect alterations of reward processing per se, but could rather correspond to an imbalance between energy supply and physical activity-mediated expenditure, which is a typical feature of EDs[43]. Therefore, we next asked whether repeatedly manipulating D1- and D2-neurons activity – as a proxy for pathological dysregulation of these neuronal subpopulation - could be sufficient for the development of some obesity- or AN-like phenotypes. To do so, mice were housed individually with free access to high fat diet and a running wheel in order to evaluate the relative preference between the two rewards. CNO was administered every day at the onset of the dark period for 3 weeks (Fig. 4A).

Chronic D2-neurons activation (D2 Gq) transiently increased food consumption as compared to WT littermate controls treated with CNO (Fig. 4B). This was accompanied by decreased voluntary wheel running throughout the entire period of CNO administration compared to controls (Fig. 4C, SF4A). These resulted in a significantly higher gain of fat mass measured by TD-NMR at the end of the chronic CNO period (Fig. 4D, SF4B-E). Importantly, cessation of CNO administration led to an immediate reversion of voluntary physical activity, together with a significant decrease in food consumption (Fig. 4B, C, SF4A), resulting in significant weight and fat mass loss compared to controls (Fig. 4D, SF4B-F). Despite the acute increase in wheel running and decrease in food consumption, as previously found (Fig. 3), animals subjected to chronic inhibition of D2-neurons (D2 Gi) did not significantly differ from controls for most of the main variables measured (Fig. 4B–D, SF4A-F) but displayed significantly higher fat gain (Fig. 4D, SF4C) that might be due to a trend of higher food consumption throughout the chronic manipulation (Fig. 4B).

## Enhanced voluntary wheel running under chronic activation of D1-neurons is compensated by increased food consumption

Despite the strong satietogenic effect of D1-neurons activation (D1 Gq) the very first day of CNO administration, the higher voluntary

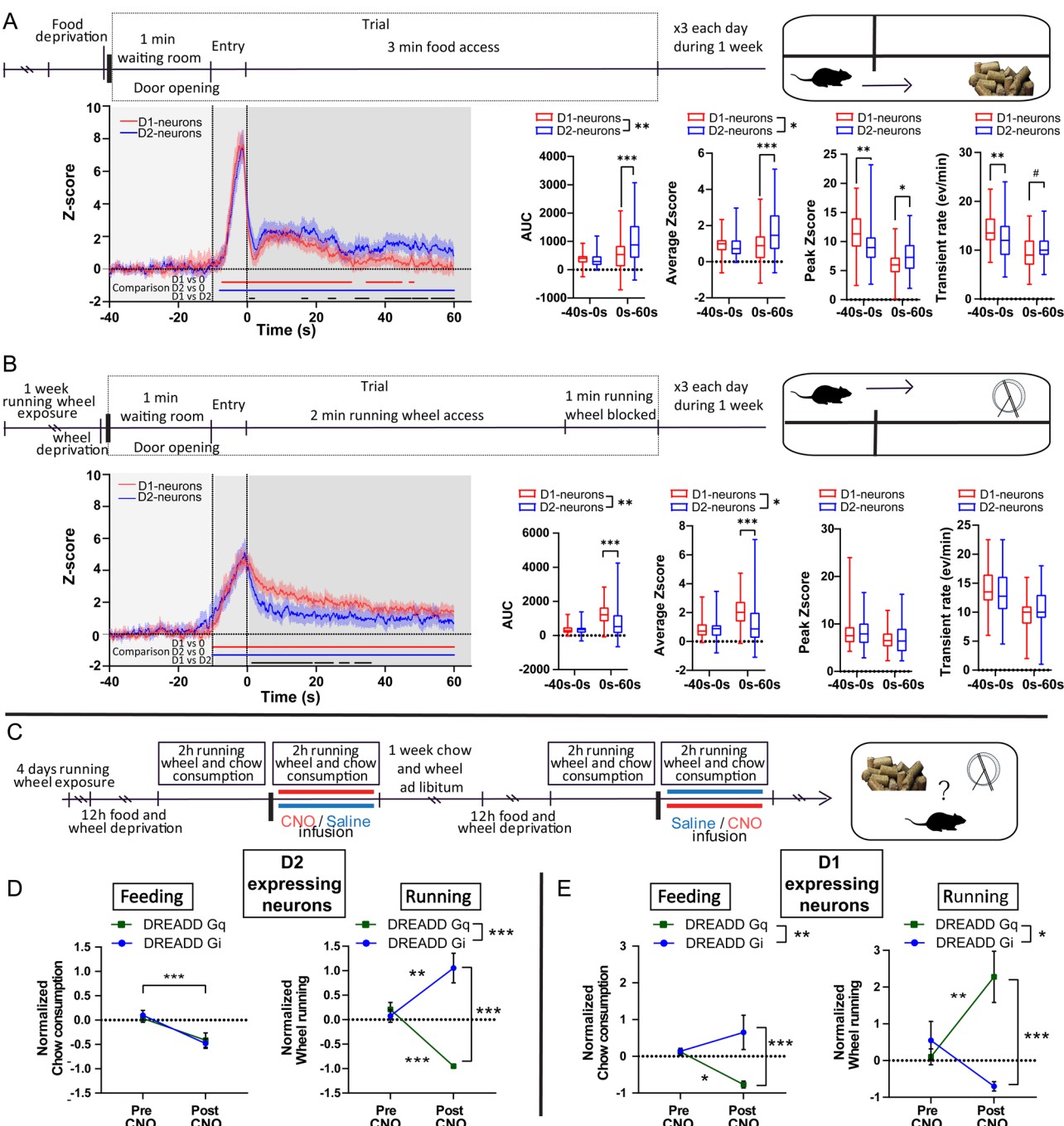

Fig. 3 | **D1- and D2-neurons of the NAc are differentially activated during feeding and running and oppositely regulate the balance between feeding and voluntary running. A** (Top) Experimental timeline for recording of calcium dynamics during feeding. (Bottom) Normalized calcium activity of D1 and D2-neurons during feeding trials (Z score; 12 trials/animal; D1 events=59; D2 events=60) with quantification of the AUC (Two-way RM ANOVA; Group effect $F_{(1, 117)} = 7.376$, $p = 0.0076$; Post-hoc D1 vs D2 0s-60s $p = 0.0001$), average z-score (Two-way RM ANOVA; Group effect $F_{(1, 117)} = 5.489$, $p = 0.0208$; Post-hoc D1 vs D2 0s-60s: $p = 0.0001$), Peak z-score (Two-way RM ANOVA; Interaction $F_{(1, 117)} = 31.07$, $p = 0.0001$; Post-hoc D1 vs D2 −40s-0s $p = 0.0099$; Post-hoc D1 vs D2 0s-60s $p = 0.0391$) and transient rates (Two-way RM ANOVA; Interaction $F_{(1, 117)} = 12.67$, $p = 0.0005$; Post-hoc D1 vs D2 −40s-0s $p = 0.0089$; Post-hoc D1 vs D2 0s-60s $p = 0.0634$).**B** (Top) Experimental timeline for recording of calcium dynamics during running. (Bottom) Normalized calcium activity of D1 and D2-neurons during running trials (Z score; 12 trials/animal; D1 events = 60; D2 events = 59) with quantification of the AUC (Two-way RM ANOVA; Group effect $F_{(1, 117)} = 7.219$, $p = 0.0083$; Post-hoc D1 vs D2 0s-60s: $p = 0.0001$), average z-score (Two-way RM ANOVA; Group effect $F_{(1, 117)} = 5.598$, $p = 0.0196$; Post-hoc D1 vs D2 0s-60s:

$p = 0.0001$), Peak z-score and transient rates. For both A and B, D2-cre: $n = 5$; D1-cre: $n = 5$; Blue and red bars represent statistical difference to baseline, black bar significant difference between the relative activity of both subpopulations. **C**–**E** (C) Experimental design. Normalized chow consumption and wheel running comparing 1-hour pre-CNO and 1-hour post-CNO injection for chemogenetic activation and inhibition of D2-neurons (D; Two-way RM ANOVA; Feeding: Infusion effect $F_{(1, 13)} = 20.70$, $p = 0.0005$; Running: Virus effect $F_{(1, 13)} = 25.27$, $p = 0.0002$; Interaction $F_{(1, 13)} = 48.01$, $p = 0.0001$; Post-hoc D2Gi vs D2Gq Post infusion $p = 0.0001$; Post-hoc Pre infusion vs Post infusion D2Gi $p = 0.0016$ D2Gq $p = 0.0002$) and D1-neurons (E; Two-way RM ANOVA; Feeding: Virus effect $F_{(1, 14)} = 9.012$, $p = 0.0095$; Interaction $F_{(1, 14)} = 8.166$, $p = 0.0127$; Post-hoc D1Gi vs D1Gq Post infusion $p = 0.0006$; Post-hoc Pre infusion vs Post infusion D1Gq $p = 0.0495$; Running: Virus effect $F_{(1, 14)} = 7.508$, $p = 0.0160$; Interaction $F_{(1, 14)} = 15.14$, $p = 0.0016$; Post-hoc D1Gi vs D1Gq Post infusion $p = 0.0001$; Post-hoc Pre infusion vs Post infusion D1Gq $p = 0.0072$). D2-cre: DREADD Gi $n = 7$, DREADD Gq $n = 7$; D1-cre: DREADD Gi $n = 8$, DREADD Gq $n = 8$. #: $0.05 < p < 0.1$; *: $p < 0.05$; **: $p < 0.01$; ***: $p < 0.001$; Error bars = s.e.m. Detailed statistics are displayed in Supplementary Table 2. Source data are provided as a Source Data file.

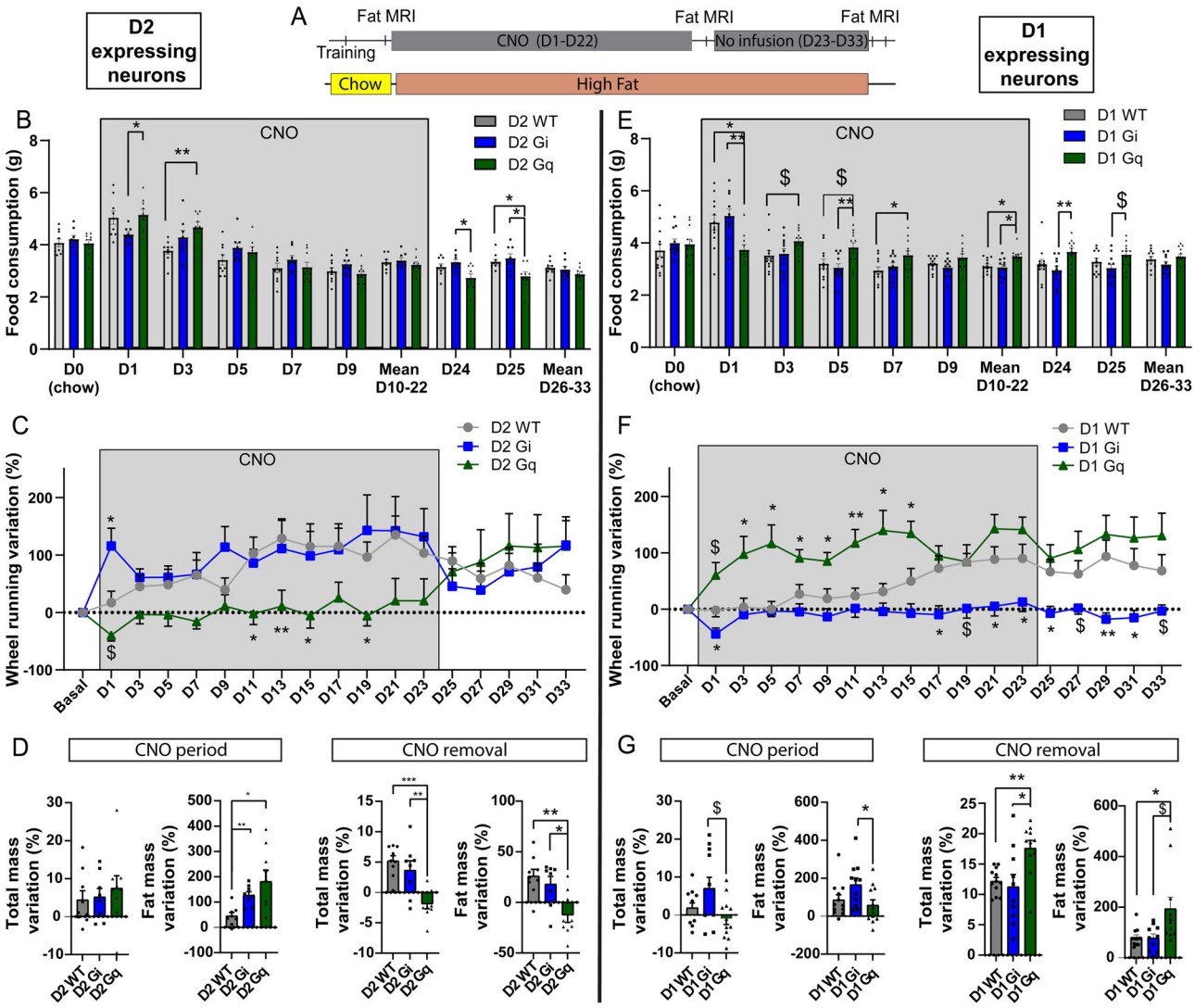

**Fig. 4 | Effect of chronic chemogenetic manipulations of D2-neurons and D1-neurons of the NAc in a wheel-running/high-fat free-feeding choice procedure.**
**A** Experimental design. **B–D** Food consumption over time (B; Two-way RM ANOVA; Interaction $F_{(18, 207)} = 4.489$, $p = 0.0001$; Post-hoc D2 Gq vs D2 WT: Day 3 $p = 0.0091$, Day25 $p = 0.0330$; Post-hoc D2 Gq vs D2 Gi: Day 1 $p = 0.0256$, Day24 $p = 0.0197$, Day25 $p = 0.0312$), variation in wheel running (C; Two-way RM ANOVA; Interaction $F_{(34, 318)} = 3.536$, $p = 0.0001$; Post-hoc D2 Gi vs D2 WT: Day1 $p = 0.0272$; Post-hoc D2 Gq vs D2 WT: Day11 $p = 0.0305$, Day13 $p = 0.0038$, Day15 $p = 0.0208$, Day19 $p = 0.0328$, Day21 $p = 0.0145$, Day23 $p = 0.0228$) and variation in total and fat mass (D; One way ANOVA; Fat Mass variation CNO period $F_{(2.00, 10.99)} = 5.966$, $p = 0.0176$; Post-hoc D1 WT vs D1 Gi $p = 0.0026$; Post-hoc D1 WT vs D1 Gq $p = 0.0447$; Total mass variation CNO removal $F_{(2, 23)} = 12.67$, $p = 0.0002$; Post-hoc D2 WT vs D2 Gq $p = 0.0002$; Post-hoc D2 Gi vs D2 Gq $p = 0.0040$; Fat Mass variation CNO removal $F_{(2, 23)} = 8.537$, $p = 0.0017$; Post-hoc D2 WT vs D2 Gq $p = 0.0020$; Post-hoc D2 Gi vs D2 Gq $p = 0.0184$) were measured under chronic D2-neurons inhibition (D2 Gi) and activation (D2 Gq) compared to WT controls, as well as after CNO cessation.
**E–G** Food consumption over time (E; Two-way RM ANOVA; Interaction

$F_{(14, 222)} = 6.702$, $p = 0.0001$; Post-hoc D1 Gq vs D1 WT: Day1 $p = 0.0183$, Day7 $p = 0.0411$; Post-hoc D1 Gq vs D1 Gi: Day1 $p = 0.0022$, Day5 $p = 0.0096$, Day24 $p = 0.0082$; One-way ANOVA mean D10-D22 $F_{(2, 32)} = 6.579$, $p = 0.0040$; Post-hoc D1 Gq vs D1 WT $p = 0.0152$, D1 Gq vs D1 Gi $p = 0.0084$), variation in wheel running (F; Two-way RM ANOVA; Virus effect $F_{(2, 28)} = 11.07$ $p = 0.0003$, Interaction $F_{(34, 471)} = 3.334$, $p = 0.0001$; Post-hoc D1 Gi vs D1 WT: Day1 $p = 0.0299$, Day17 $p = 0.0187$, Day21 $p = 0.0174$, Day25 $p = 0.0223$, Day29 $p = 0.009$, Day31 $p = 0.0325$; Post-hoc D1 Gq vs D1 WT: Day3 $p = 0.0223$, Day5 $p = 0.0129$, Day7 $p = 0.0218$, Day9 $p = 0.0213$, Day11 $p = 0.0086$, Day13 $p = 0.0290$, Day15 $p = 0.0285$) and variation in total and fat mass (**G**; One way ANOVA; Total mass variation CNO period $F_{(2.00, 20.08)} = 3.861$, $p = 0.0381$; Post-hoc D1 Gi vs D1 Gq $p = 0.0826$; Fat Mass variation CNO period $F_{(2, 32)} = 3.492$, $p = 0.0425$; Post-hoc D1 Gi vs D1 Gq $p = 0.0463$) were measured under chronic D1-neurons inhibition (D1 Gi) and activation (D2 Gq) compared to WT controls, as well as after CNO cessation. D2 WT $n = 9$; D2 Gi $n = 8$; D2 Gq $n = 9$; D1 WT $n = 12$; D1 Gi $n = 11$; D1 Gq $n = 12$. $: $0.05 < p < 0.1$; *: $p < 0.05$; **: $p < 0.01$; Error bars = s.e.m. Detailed statistics are displayed in Supplementary Table 2. Source data are provided as a Source Data file.

physical activity that was maintained across chronic D1-neurons activation was accompanied by higher food consumption during the entire period of CNO administration compared to controls (Fig. 4E, F, SF4G). As a consequence, animals with chronic activation of D1-neurons did not differ from controls in terms of body weight, fat, and lean mass variation (Fig. 4G, SF4H-L), suggesting that energy expenditure resulting from increased physical activity was compensated by higher energetic supply. Cessation of CNO administration progressively normalized both wheel running and food intake

relative to controls, leading to significant gain in fat mass in the D1 Gq group (Fig. 4G, SF4H-L).

Chronic inhibition of D1-neurons (D1 Gi) blunted wheel running, with no difference in food consumption compared to controls (Fig. 4E, F, SF4G). This resulted in significant weight gain and increased fat mass as compared to the D1 Gq group (Fig. 4G, SF4H-L). Interestingly, lower voluntary physical activity was maintained despite cessation of CNO administration (Fig. 4F, SF4G) with no change in body weight or fat mass (Fig. 4G, SF4H-L).

## Concomitant chronic manipulation of both NAc subpopulations bypasses the equilibrium between activity-related energy expenditure and energy intake

Altogether, our findings suggest that decreased energy expenditure due to lower voluntary wheel running, as observed under activation of D2-neurons or inhibition of D1-neurons, does not lead to compensatory decrease in food intake. Such an alteration of energy balance leads to fat gain, i.e. an obesity-like phenotype. This is in line with previous observation consecutive to decreased expression of the D2 receptor[36,37] – which disinhibits D2-neurons. However, enhanced running-induced energy expenditure, as we found under D1-neurons activation led to compensatory increase in food intake, thus maintaining energy balance, as reflected by the maintenance of weight and fat mass. We therefore asked whether potentiating the disequilibrium between D1- and D2-neuron activities by manipulating both subpopulations concomitantly in opposite manner, could heighten the aforementioned phenotypes, eventually leading to bidirectional disruptions of the balance between food intake and activity-induced energy expenditure. We took advantage of a unique mouse model expressing the flippase recombinase in D1-expressing neurons[44] and crossed this mouse line with the D2-cre line, thereby allowing the concomitant manipulation of both NAc subpopulations (Fig. 5A). Simultaneous activation of D1-neurons together with inhibition of D2-neurons strongly increased wheel running. However, in this case, after an acute decrease the first day of CNO administration, food consumption remained equivalent to WT control animals (Fig. 5B, C,

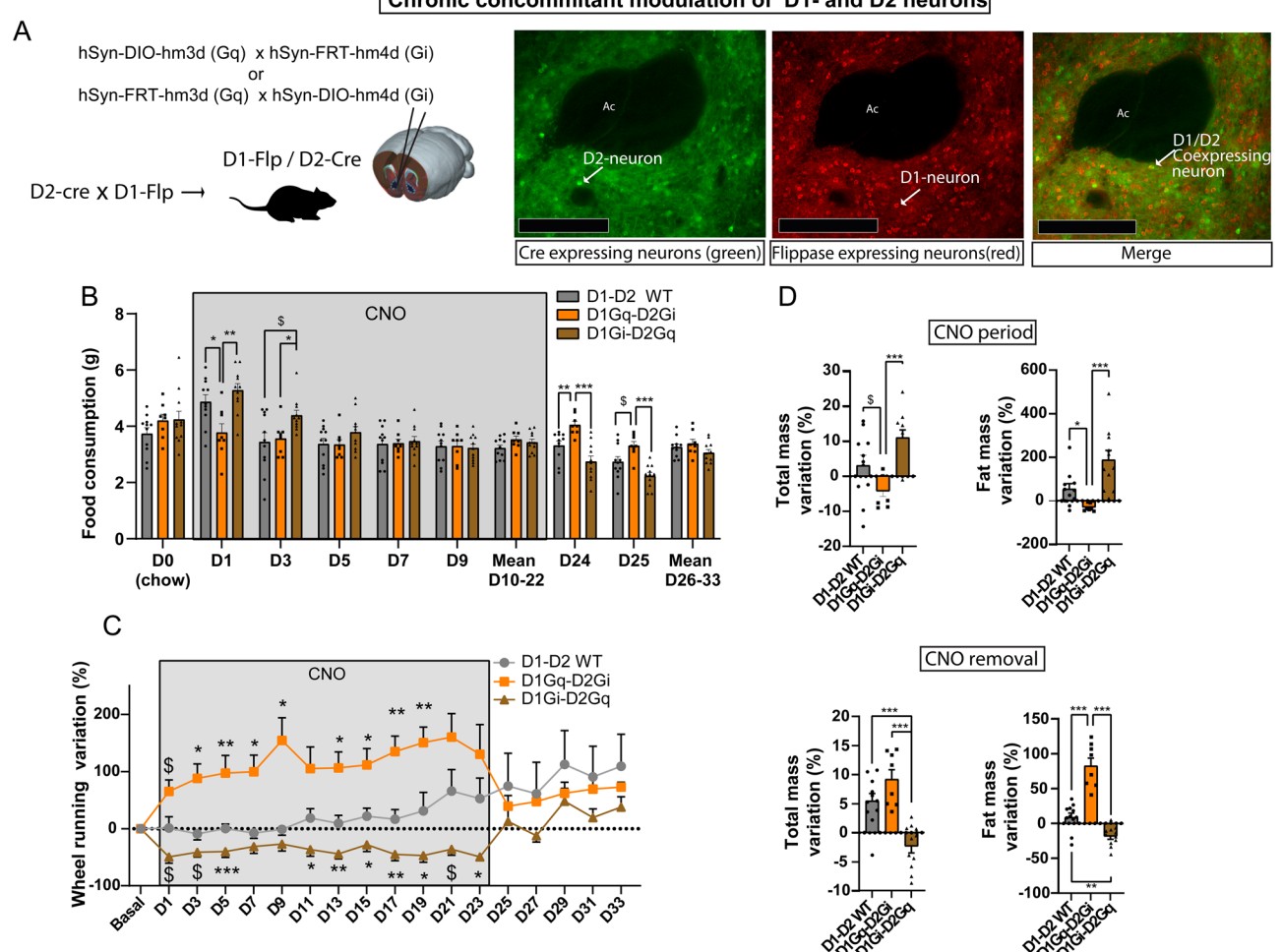

**Fig. 5 | Effect of chronic and simultaneous chemogenetic manipulations of D2-neurons and D1-neurons of the NAc in a wheel-running/high-fat free-feeding choice procedure. A** Schematic design for the transgenic and viral strategy for concomitant chemogenetic manipulation of D1- and D2-neuroins of the NAc (left) and representative pattern of expression of cre-dependent GFP and flp-dependent mCherry in double transgenic D1-FlpxD2-cre mice (right). Scale bar 250 μm. **B–D** Food consumption over time (**B**; Two-way RM ANOVA; Interaction $F_{(14, 189)} = 7.911$, $p = 0.0001$; Post-hoc WT vs D1Gq-D2Gi: Day1 $p = 0.0498$, Day24 $p = 0.0056$, Day25 $p = 0.0769$; Post-hoc WT vs D1Gi-D2Gq: Day3 $p = 0.0648$, Day24 $p = 0.0967$; Post-hoc D1Gi-D2Gq vs D1Gq-D2Gi: Day1 $p = 0.0057$, Day3 $p = 0.0277$, Day24 $p = 0.0004$, Day25 $p = 0.0001$), variation in wheel running (**C**; Two-way RM ANOVA; Virus effect $F_{(2, 23)} = 8.839$ $p = 0.0014$; Interaction F(34, 376) = 3.486, $p = 0.0001$; Post-hoc WT vs D1Gq-D2Gi: Day1 $p = 0.0791$, Day3 $p = 0.0122$, Day5 $p = 0.0013$, Day7 $p = 0.0148$, Day9 $p = 0.0105$, Day13 $p = 0.0106$, Day15 $p = 0.0164$, Day17 $p = 0.0059$, Day19 $p = 0.0010$; Post-hoc D1Gi-D2Gq vs WT: Day1 $p = 0.0966$, Day3 $p = 0.0600$, Day5 $p = 0.0005$, Day11 $p = 0.0336$, Day13 $p = 0.0039$, Day15

$p = 0.0331$, Day17 $p = 0.0047$, Day19 $p = 0.0282$, Day21 $p = 0.0568$, Day23 $p = 0.0443$) and variation in total and fat mass (**D**; One way ANOVA; Total mass variation CNO period $F_{(2, 27)} = 9.421$, $p = 0.0008$; Post-hoc WT vs D1Gi-D2Gq $p = 0.0638$; Post-hoc D1Gq-D2Gi vs D1Gi-D2Gq $p = 0.0006$; Fat mass variation CNO period Kruskal-Wallis $p = 0.0002$; Post-hoc WT vs D1Gq-D2Gi $p = 0.0408$; Post-hoc D1Gq-D2Gi vs D1Gi-D2Gq $p = 0.0001$; Total mass variation CNO removal $F_{(2, 27)} = 19.79$ $p = 0.0001$; Post-hoc WT vs D1Gi-D2Gq $p = 0.0004$; Post-hoc D1Gq-D2Gi vs D1Gi-D2Gq $p = 0.0001$; Fat mass variation CNO removal $F_{(2, 27)} = 52.97$ $p = 0.0001$; Post-hoc WT vs D1Gi-D2Gq $p = 0.0171$; Post-hoc WT vs D1Gq-D2Gi $p = 0.0001$; Post-hoc D1Gq-D2Gi vs D1Gi-D2Gq $p = 0.0001$) were measured under chronic manipulations of both subpopulations (D1Gq-D2Gi: concomitant activation of D1-neurons and inhibition of D2-neurons; D1Gi-D2Gq: concomitant inhibition of D1-neurons and activation of D2-neurons) compared to WT controls, as well as after CNO cessation. D1-D2 WT $n = 9$; D1Gi-D2Gq $n = 9$, D1Gq-D2Gi $n = 8$. \$: $0.05 < p < 0.1$ *: $p < 0.05$; **: $p < 0.01$; ***: $p < 0.001$ Error bars = s.e.m. Detailed statistics are displayed in Supplementary Table 2. Source data are provided as a Source Data file.

SF5A), preventing weight and fat gain (Fig. 5D, SF5B-F). In opposite, the concomitant inhibition of D1-neurons together with activation of D2-neurons resulted in significantly lower wheel running compared to controls, together with an initial increase in food consumption over the first 3 days, that returned to control levels thereafter (Fig. 5B, C, SF5A). Such a behavioral pattern resulted in higher weight gain and increased fat mass (Fig. 5D, SF5B-F). Cessation of CNO administration normalized voluntary physical activity in both groups (Fig. 5C, SF5A). Interestingly, this was accompanied by an increase of food intake (Fig. 5B) and fat gain in animals in which D1- and D2-neurons were previously activated and inhibited, respectively (Fig. 5D, SF5B-F). Conversely, food intake (Fig. 5B) and body weight (Fig. 5D, SF5B-F) were decreased in animals in which D1- and D2-neurons were previously simultaneously inhibited and activated, respectively.

## Chronic imbalance of D1- and D2-neuron activities in favor of D1-neurons leads to anorexia-like phenotype

The latter findings show that the effect of chronic concomitant manipulation of D1- and D2-neurons on voluntary activity can overcome metabolic signals related to energy intake. In particular, imbalance of activity between D1- and D2-neurons in favor of D1-neurons of

the NAc lead to weight loss due to disproportional voluntary exercise with regard to food intake, a main feature of AN. Therefore, we next asked to which extend such manipulation of NAc subpopulation could lead to pathological weight loss (up to 20%). We first exposed mice to a gradual food deprivation in the presence of the running wheel. Under such conditions, all groups displayed a decrease in weight that was accelerated under concomitant activation of D1-neurons and inhibition of D2-neurons due to higher voluntary wheel running despite food restriction (Fig. 6A, B). Strikingly, in these animals, serum level of the anorexigenic adipokine leptin was decreased while the orexigenic gut peptide ghrelin was elevated (Fig. 6C), which is a hallmark of AN[45]. Nonetheless, in the absence of a running wheel, concomitant manipulations of NAc subpopulations did not prevent the ability of leptin and ghrelin to exert their respective anorexigenic and orexigenic influence, despite that the effect of chemogenetic manipulations of NAc subpopulations was dominant on food intake (SF6A-B). These findings support that imbalanced activity of NAc subpopulations in favor of D1-neurons can overcome peripheral feeding signals.

To further support the translational validity of our findings, we subjected animals to the Activity-Based Anorexia (ABA) model[46]. Concomitant activation of D1-neurons together with inhibition of

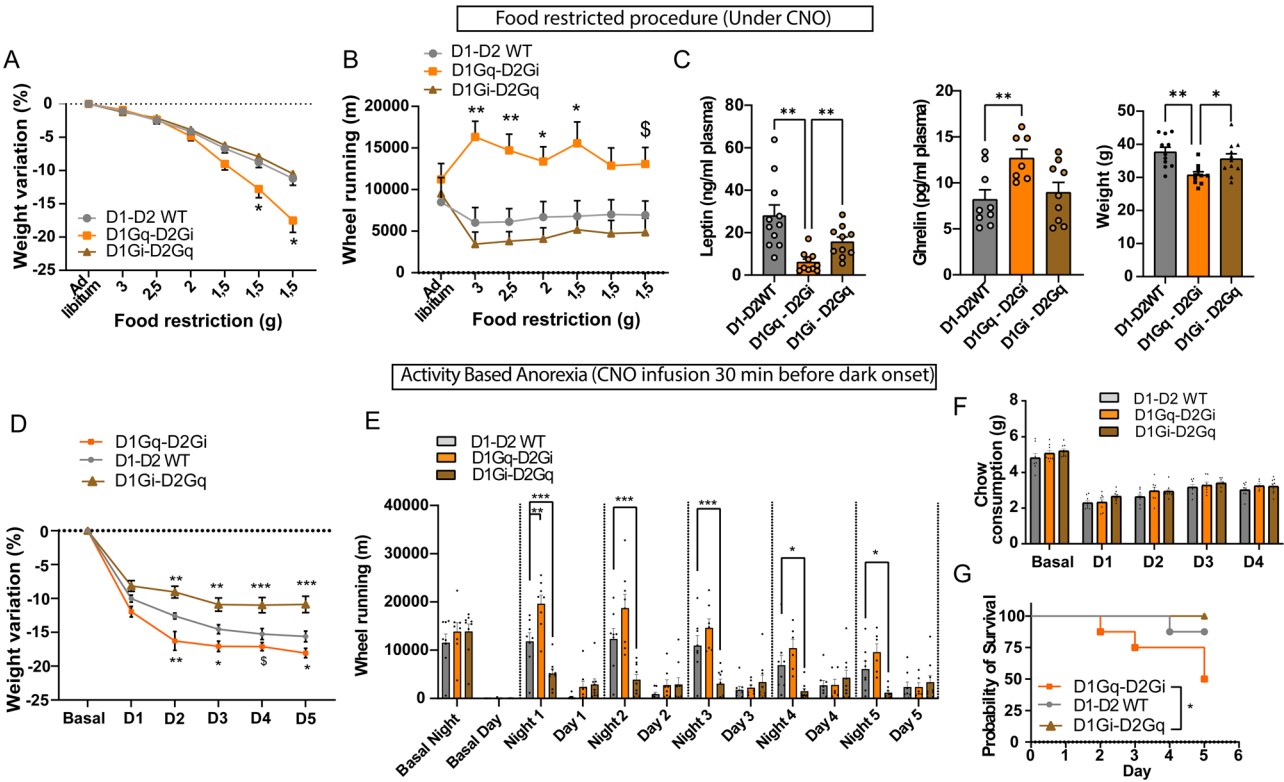

**Fig. 6 | Concomitant activation of D1- and inhibition of D2-neurons precipitates weight loss in rodent models of anorexia nervosa. A, B** Measures of weight variation (**A**; Two-way RM ANOVA; Virus effect $F_{(2, 27)} = 4.600$, $p = 0.0191$; Interaction $F_{(12, 162)} = 10.63$, $p = 0.0001$; Post-hoc WT vs D1Gq-D2Gi: 1.5 g (Day 5) $p = 0.0467$, 1.5 g (Day 6) $p = 0.0212$) and wheel-running (**B**; Two-way RM ANOVA; Virus effect $F_{(2, 23)} = 7.484$, $p = 0.0031$; Interaction $F_{(12, 136)} = 5.156$, $p = 0.0001$; Post-hoc WT vs D1Gq-D2Gi: 3 g $p = 0.0027$, 2.5 g $p = 0.0085$, 2 g $p = 0.0429$, 1.5 g (Day 5) $p = 0.0318$, 1.5 g (Day 7) $p = 0.0648$) in a procedure of progressive food restriction. **C** Dosage of plasma leptin and ghrelin levels and weight following the food-restriction procedure (One way ANOVA; Leptin $F_{(2.00, 15.65)} = 10.99$, $p = 0.0010$; Post-hoc WT vs D1Gq-D2Gi $p = 0.0038$; Post-hoc WT vs D1Gi-D2Gq $p = 0.1179$; Post-hoc D1Gq-D2Gi vs D1Gi-D2Gq $p = 0.0088$; Ghrelin $F_{(2, 22)} = 4.870$, $p = 0.0177$; Post-hoc WT vs D1Gi-D2Gq $p = 0.0142$; Weight $F_{(2, 29)} = 7.048$, $p = 0.0032$; Post-hoc WT vs D1Gi-D2Gq $p = 0.0029$; Post-hoc D1Gq-D2Gi vs D1Gi-D2Gq $p = 0.0463$. **D–G** Measures of weight variation (**D**; Two-way RM ANOVA; group effect $F_{(2, 22)} = 17.45$, $p = 0.0001$;

Interaction $F_{(10, 105)} = 7.702$, $p = 0.0001$; Post-hoc D1-D2WT vs D1Gq-D2Gi: D2 $p = 0.0046$, D3 $p = 0.0312$, D5 $p = 0.0220$; Post-hoc D1-D2WT vs D1GI-D2Gq: D2 $p = 0.0051$, D3 $p = 0.0038$, D4 $p = 0.0006$, D5 $p = 0.0001$), wheel running (**E**; Two-way RM ANOVA; group effect $F_{(2, 22)} = 7.078$, $p = 0.0042$; Interaction $F_{(22, 232)} = 9.935$, $p = 0.0001$; Post-hoc D1-D2WT vs D1Gq-D2Gi: Night 1 $p = 0.0004$, Night 2 $p = 0.0051$; Post-hoc D1-D2WT vs D1Gi-D2Gq: Night 1 $p = 0.0023$, Night 2 $p = 0.0001$, Night 3 $p = 0.0001$, Night 4 $p = 0.0190$, Night 5 $p = 0.0381$) and chow consumption (**F**) in an Activity-Based Anorexia (ABA) model. **G** Probability of survival (i.e. % of animals displaying less than 20% weight loss) over the ABA procedure (Log rank test: D1Gi-D2Gq vs D1Gq-D2Gi χ2 = 5.583, $p = 0.0189$). Food restriction and dosage: D1-D2 WT $n = 9$, D1Gi-D2Gq $n = 9$, D1Gq-D2Gi $n = 8$; ABA: D1-D2 WT $n = 8$, D1Gi-D2Gq $n = 9$, D1Gq-D2Gi $n = 8$. \$: $0.05 < p < 0.1$ *: $p < 0.05$; **: $p < 0.01$; ***: $p < 0.001$ Error bars = s.e.m. Detailed statistics are displayed in Supplementary Table 2. Source data are provided as a Source Data file.

D2-neurons precipitated weight loss due to higher voluntary activity (Fig. 6D–F, SF6C-D), leading to lower probability of "survival" (20% weight loss, Fig. 6G). Concomitant inhibition of D1-neurons together with activation of D2-neurons was "protective" due to blunted voluntary running (Fig. 6D–G, SF 6C-D).

## Discussion

Beyond the well-documented role of NAc in the modulation of incentive processes, the present study highlights an underestimated differential implication of D1- and D2-neuronal subpopulations in the balance between food intake and physical activity-induced energy expenditure. As an illustration, the use of the above-mentioned double transgenic mice reveals that repeated (i) activation of D2-neurons together with (ii) inhibition of D1-neurons is sufficient to elicit an obesity-like phenotype as assessed by a decrease in energy expenditure. In opposite, repeated hyperactivity of D1-neurons together with hypoactivity of D2-neurons directs the energetic balance towards physical activity at the expense of energy intake, which are typical features of AN.

Even though dimensional approaches to EDs is just starting to emerge, increasing evidence supports alterations in common domains across EDs, especially those linked to reward-related positive valence system[5]. In particular, reward valuation is elevated in individuals with obesity-related binge eating syndromes, but excessively diminished in patients with AN[47,48]. Similarly, increased willingness to work for disorder-specific reward, i.e. exercise and food for AN and bulimia, respectively, are increased[49,50]. Such reward-related symptoms correlate with opposite brain activity patterns, especially in cortico-striatal circuits[13], hence reinforcing the hypothesis of shared neurobiological substrates, and therefore potential continuum in EDs. Our data suggest that opposite alterations in the activity of NAc subcircuits could underlie a subset of symptomatic dimensions of ED phenotypes, in particular the balance between calorie intake and activity-related energy expenditure.

We found that the bulk activity balance between D1- and D2-neuron subpopulations was in favor of D2-neurons during food intake while towards D1-neurons during voluntary physical exercise. In accordance, acute manipulations of the two subpopulations led to opposite consequences on food intake and voluntary physical activity, highlighting the implication of these neuronal subpopulations in the regulation of energy balance. However, it is important to stress that, under chronic manipulations, the homeostatic system – likely comprising hypothalamic network - seems to be able to overcome alteration of D1- and D2-neurons' activity. Thus, under chronic activation of D1-neurons, despite an acute decrease in food intake, enhanced energy expenditure through physical activity is accompanied by increased food intake, allowing maintenance of fat mass and body weight. These findings are mostly consistent with a previous study by the group of DiLeone[31]. However, our findings reveal that it is only when both NAc subpopulations are concomitantly manipulated, i.e. activation and inhibition of D1- and D2-neurons, respectively, that such an homeostatic system does not fully compensate for energy expenditure, hence leading to weight loss. Nonetheless, animals still eat comparable amount of food than controls, suggesting that the energetic imbalance mainly results from exaggerated physical activity-mediated energy expenditure rather than reduced food consumption. Similarly, when D1-neurons or D2-neurons are respectively subjected to chronic inhibition and activation, weight/fat gain mostly originates from decreased physical activity since animals consume comparable amount of food than controls. Finally, under food restriction (i.e. progressive food restriction or ABA model), we found that protection from, or precipitation of, weight loss depending on the manipulations of dopaminoceptive subpopulations, mainly resulted from alterations of voluntary activity. Yet, avoidance and compulsion towards exercise are typical features of obesity-related disorders and AN, respectively,

that have been proposed to be one main pathophysiological mechanism leading to, and even preceding, the development of EDs[6,51]. Even though speculative, overall, our findings suggest that chronic dysfunction in the balance of activities of D1- and D2-neurons of the NAc could participate to the etiology of EDs by altering, positively or negatively, the willingness to engage in, and/or the rewarding value of, physical activity. These findings expand the importance of the balance of activities between striatal D1- and D2-neurons to control behavior[52,53].

Of importance is the observation that wheel-running, rather than reflecting non-specific locomotor responses, is a goal-directed behavior accounted for by the reinforcing value of exercise. Thus, as for other natural rewards such as food, rats/mice are willing to exert efforts, i.e. lever presses or nose pokes, to access a wheel[54,55]. In accordance, our findings using chemogenetic inhibition demonstrate that D1-neurons facilitate both wheel-running and willingness to exert effort to obtain a food reward, while D2-neurons rather play an opposite role. These data further suggest that alterations in voluntary physical activity under D1- and D2-neurons manipulation reflect changes in reward/incentive processes and reveal the antagonistic role of the two NAc subpopulations on this dimension. However, such an opposite effect of D1- and D2-neurons on motivational processes has been recently challenged. Similarly to our findings, the chemogenetic inhibition of D2-neurons of the NAc enhances performance in a motivational task[21,22], while optogenetic manipulations of D2-neurons of the NAc rather support a "pro-motivational" role, similarly to D1-neurons[23–25]. However, recent data clarified this discrepancy, showing that optogenetic stimulation of D2-neurons during reward-predicting cue enhances operant responding, while the very same manipulation during reward consumption has the opposite effect[27]. Such a dissociated role of D2-neurons on distinct phases of the motivational dimension remains unclear and might involve reward prediction error mechanisms through dynamic feedback modulation of VTA dopamine neurons[27]. Nonetheless, through the use of long-lasting chemogenetic approach, our results support that chronic increase or decrease in D2-neurons excitability, as putatively found in some pathological conditions, will likely lead to enhanced and blunted motivational processes, respectively.

In addition to their antagonistic effect on the "ability to exert effort", our results reveal that D1-neurons inhibit, whereas D2-neurons of the NAc enhance food intake. To the best of our knowledge, such a role of D2-neurons has not been reported before. However, even though a recent study showed that activation of D1- and D2-neurons of the NAc enhance and terminate sugar intake, respectively[56], the acute anorexigenic potency of D1-neurons has been previously described for the NAc shell, through projections onto the lateral hypothalamus and the midbrain[28–30]. These latter findings raise the question of whether these two pathways are redundant to control food intake or modulate distinct – though complementary – dimensions of feeding behavior. It has been proposed that the NAc-VTA pathway modulates feeding through a direct control onto motor behaviors that comprise a feeding response[28,57]. Regarding NAc-LH projections that mostly originate from the shell of the NAc, GABA signaling – notably originating from NAc D1-expressing neurons – has been shown to be increased during food intake, consequently terminating feeding, notably through decrease of orexin production[58]. These findings also highlight that, beyond the core and shell subdivisions, subgroups of NAc D1-neurons likely control discrete behavioral dimensions, related to their location in subterritories of the NAc as well as their input and output projections. Finally, a main output structure of D1-expressing neurons of the NAc is the ventral pallidum (VP). Lesion of the VP – similarly to lesions of the LH - has been shown to counteract the orexigenic effect of NAc inhibition[59], suggesting a "pro-feeding" role of the VP. Accordingly, arkypallidal neurons of the VP potentiate reward consumption[60]. Even though, to the best of our knowledge, a direct implication of

D1-neuron-to-VP projections in the regulation of food intake remains to be demonstrated, all these findings support that D1- neurons of the NAc can decrease food intake through several output pathways involving the LH, the VTA and the VP. Of note, even though we cannot exclude some effects related to spread of viral infections to the shell of the NAc, our chemogenetic manipulations were mostly restricted to the core subdivision, that sends projections to the VP and VTA but not the LH. The circuits by which D2-neurons of the NAc regulate feeding remain unclear. As mentioned, we cannot rule out that some of the effects we observed in this study might be related to an alteration in the activity of cholinergic interneurons (CIN) due the use of the D2-cre line - even though likely targeting at most a subset of ChAT-positive interneurons[22]. However, increasing CINs activity reduces palatable food consumption while their inhibition had the opposite effect[61], in contrary to our findings with manipulations of D2-neurons. This suggests that most of the effects we observed were due to alterations of D2-MSNs. A main output of this neuronal subpopulation is the VP. This raises the intriguing idea that, if the VP is involved in the effects on feeding that we observed under manipulation of NAc subpopulations, it is likely that D1- and D2-neurons project onto distinct VP neuronal subpopulations. However, to the best of our knowledge, this is an important aspect that remains to be demonstrated. Finally, the lateral inhibition between NAc subpopulations has been largely overlooked and is likely to participate to their differential roles, including feeding behavior.

Together with the current study, these observations suggest that, under physiological conditions, the recruitment of the NAc network in favor of D1-neurons would bias behavior towards effort exertion – including food-seeking – at the expense of food consumption. In opposite, a predominant activity of D2-neurons would blunt voluntary activity whilst promoting food intake. Such a pivotal role of the NAc neuronal network is in accordance with the implication of the NAc as a "sensory sentinel", allowing rapid control over food consumption in response to motivational or sensory signals[28,29,57,62]. It remains to be determined whether these behavioral dimensions bidirectionally modulated by D1- and D2-neurons depend on distinct subpopulations in the NAc and/or their projecting areas. Nonetheless, the current work suggests that a chronic imbalance in the NAc neuronal network could be sufficient to alter homeostatic balance, eventually leading to EDs.

## Methods
Key resources and reagents are listed in Supplementary Table 1.

### Experimental models and subject details
Adult male C57BL6/J, D1-Cre, D2-Cre, D1-Flp and D1Flp/D2cre (home-made breeding from D1Flp and D2cre mice) mice were used in this study (see Supplementary Table 1 for details). Mice (12 to 28 weeks old) were group housed (4-7 animals), unless otherwise specified, in standard polypropylene cages and maintained in a temperature and humidity-controlled environment under a 12:12 light-dark cycle (07:30 on) with *ad libitum* access to water and food. For food consumption experiments, mice were housed in standard polypropylene individual cages. All behavioral procedures were carried out during the light phase unless otherwise specified. Mice were habituated to handling prior to behavioral procedures, and to injection proceeding before test day. All animal care and experimental procedures were in accordance with the INRAE Quality Reference System and to French legislations (Directive87/148, Ministère de l'Agriculture et de la Pêche) and European (Directive 86/609/EEC). They followed ethical protocols approved by the Region Aquitaine Veterinary Services (Direction Départementale de la Protection des Animaux, approval ID: B33-063-920) and by the animal ethic committee of Bordeaux CEEA50. Every effort was made to minimize suffering and reduce the number of animals used.

### Stereotaxic viral injections and implantation of optic fibers
Mice analgesia was achieved with the subcutaneous injection of buprenorphine (0.05 mg/kg). The animals were then anesthetized using isoflurane (4% induction, 1.5% maintenance) and placed in a stereotaxic frame onto a heating pad. The AAV8-hSyn-DIO-hM4D-mCherry virus (titer of $4.5 \times 10^{12}$ vg/ml l) was used for the Cre-dependent expression of hM4D(Gi) receptor, the AAV8-hSyn-DIO-hM3D-mCherry virus (titer of $5.2 \times 10^{12}$ vg/ml) was used for the Cre-dependent expression of hM3D(Gq) receptors and the AAV8-hSyn-DIO-mCherry virus ($1 \times 10^{13}$ vg/ml) was used for the cre dependent expression of mCherry. The AAV8-hSyn1-dFRT- mCherry(rev)-dFRT-WPRE-hGHp(A) and AAV8-hSyn1-dlox-EGFP(rev)-dlox-WPRE-hGHp(A) were used for the labeling of D1-Flp and D2-cre expressing neurons (titer of $4-5 \times 10^{12}$ vg/ml). The AAV8-hSyn1-dFRT-hM4D(Gi)-mCherry(rev)-dFRT-WPRE-hGHp(A) and AAV8-hSyn1-dFRT-hM3D(Gq)-mCherry(rev)-dFRT-WPRE-hGHp(A), were used for the flp-dependent expression of hM3D(Gq) and hM4D(Gi)-receptors (titer of $4 \times 10^{12}$ vg/ml). A total of 0.5 μl of virus solution was injected bilaterally in the nucleus accumbens using a 10 μl Hamilton syringe at a rate of 100 nl/mn at the following stereotaxic coordinates relative to Bregma (in mm): Ap = +1,7; ML = +/− 1,1; DV = −4 from the surface of the brain. The syringe was left in place for 5 min before being slowly removed. For measures of calcium dynamics by fiber photometry, the AAV5-hSyn1-dlox-jGCaMP8m(rev)-dlox-WPRE-SV40p(A) viral vector was used for the cre-dependent expression of GCaMP8m ($8.0 \times 10^{12}$ vg/ml). For the measure of dopamine dynamics coupled to chemogenetic activation of D1- or D2-neurons, the AAV9-hSyn1-chl-dLight1.3b-WPRE-bGHp(A) vector was co-injected in the NAc of D1- or D2-cre mice together with the AAV8-hSyn-DIO-hM3D-mCherry vector. An optic fiber (RWD Fiber Optic Cannulae 0,5 NA, 400 μm flat tip, L:6 mm) was implanted just above the viral vector infusion site (DV = −3,75) and immobilized with an opaque dental cement. The skin incision was sutured, post-operative analgesia was prolonged with subcutaneous injections, and mice were placed on a heating pad until full recovery from the anesthesia. Behavioral experiments were started 3 to 4 weeks after the surgeries.

### Ventral tegmental area recordings
Stereotaxic surgery for electrophysiology experiments was performed under isoflurane anesthesia[63]. Recording pipettes were inserted into the VTA with the skull flat, at the following coordinates: −3.16 mm from bregma; 0.5 mm from midline. A glass micropipette (tip diameter = 2–3 μm, 4–6Mohm) filled with a 2% pontamine sky blue solution in 0.5 M sodium acetate was lowered into the VTA. DA neurons were identified according to well-established electrophysiological features[64]. These included: 1) an half action potential width ≥1.1 ms; 2) slow spontaneous firing rate (<10 Hz); 3) single and burst spontaneous firing patterns. Through these electrodes, the extracellular potential was recorded with an Axoclamp2B amplifier in the bridge mode. The extracellular potential amplified 10 times by the Axoclamp2B amplifier was further amplified 100 times and filtered (low-pass filter at 300 Hz and high-pass filter at 0.5 kHz) via a differential AC amplifier (model 1700; A-M Systems, Carlsborg, WA). Single neuron spikes were discriminated and digital pulses were collected on-line using a laboratory interface and software (CED 1401, SPIKE 2, Cambridge Electronic Design). Four parameters of VTA DA neuron impulse activity were computed over 200-second epochs after a 5-minute stable baseline period: 1) the basal firing rate, 2) the basal bursting rate, 3) the mean number of spikes per bursts and 4) the number of spontaneously active cells per track. The onset of a burst was defined as the occurrence of two spikes with an interspike interval <80msec[64]. VTA DA neurons were recorded under saline or CNO injection (2 mg/kg) in D1- or D2-cre mice expressing the cre-dependent hM3D(Gq) DREADDs in the NAc.

## Behavioral procedures

For most behavioral procedures, two experimenters were involved, one in charge of injections (CNO/saline), the other performing the behavioral tasks. The latter was blind of animals' genotype, viral vector and injection. For all behavioral experiments, mice received CNO (0.5 or 2 mg/kg dissolved in sterile 0.9% saline as indicated in the text; see Supplementary Table 2 for details), JHU37160 (1 mg/kg dissolved in sterile 0.9% saline), or 0,9% saline solution intraperitoneal (IP) injections 30 min before the beginning of the test. For the experiments requiring training sessions, there were no injections during these steps. CNO and JHU37160 solutions were prepared daily and protected from light. For acute CNO treatment experiments, each mouse received either CNO or saline in a counterbalanced manner for within subject analysis, across 2 sessions of each behavioral test at minimum. The same procedure was applied for experiments using JHU37160. For experiments using WT littermate controls, all animals were treated with CNO when indicated. Experimental designs were randomized regarding genotype, virus type, home-cage and testing chamber. Different cohorts were used for each experiment, except for the measure of licking microstructures that were performed in the same animals as the ones used for operant conditioning experiments.

**Operant conditioning.** Mice were placed under food deprivation and maintained at 85-90% of their baseline *ad libitum* weight. Fourteen operant chambers (30x40x36 cm, Imetronic Bordeaux), each located into a soundproof and light-attenuating cabinet were used. They consisted of 2 plexiglas panels and 2 PVC panels on top of a metal grid floor, lit by a houselight located at the top of the chamber during sessions. A trough was located on the center of one PVC panel, 4 cm above the metal grid, and was surrounded by 2 equally distant retractable metal levers (2x4x1cm). A cue light was located 8.5 cm above each lever. The reinforcer was composed of a 15 µl drop of sweetened condensed milk (3.25 Kcal/g) diluted at 10% in tap water delivered in the trough. Both levers were extended at the beginning of each session, except for pavlovian training. Only one lever was reinforced across all operant procedures and pressing on the non-reinforced lever had no consequence. For each mouse, all of the operant procedures were executed in the same operant chamber. Operant chambers were connected to a computer equipped with POLY Software (Imetronic), which allowed for control of the sessions parameters and recording of the data (number of presses on both levers, number of licks in the full or empty trough).

**Pavlovian training preceding operant conditioning.** Three to 4 weeks after AAV injections, animals started pavlovian training. Mice were placed in the operant chamber for 30 min sessions, one session per day, during which a reward – a drop of sweetened condensed milk – was delivered in the trough every minute, associated with the emission of a sound (65 db, 3000 Hz, 200 ms) and the illumination of the diode located above the future reinforced lever. The reward had to be consumed by the animal in order for a new reward to be delivered. The learning criterion was set at a mean of 20 rewards earned out of 30, across at least 3 daily consecutive sessions.

**Fixed ratio training (FR).** When the mice reached the pavlovian training criterion, animals started operant conditioning sessions during which they had to press on the reinforced lever to obtain the reward. Reinforced lever presses were signaled by the emission of the sound and the cue light. Operant learning started with daily 1 h fixed ratio 1 schedule of reinforcement (FR1), during which each reinforced-lever press was rewarded. The learning criterion was set at 100 reward dose earned, across four consecutive sessions.

**Random Ratio Training (RR).** Once all the mice completed the operant learning criterion, their ability to press on the reinforced lever in order

to receive a reward was increased using random ratio schedules. During the sessions, each lever press on the reinforced lever had a given probability of being reinforced (pRR5 = 0.2, pRR10 = 0.1, pRR20 = 0.05). Animals underwent those 3 different ratios across one-hour sessions, each day, 5 days a week. Moving from one ratio to the next one was achieved once lever response values across two consecutive sessions were stable (less than 10% variation).

**Progressive Ratio Task (PRx2).** The motivational component of operant conditioning was evaluated using a progressive ratio task. During the session, the first reward was delivered after two lever presses on the reinforced lever and the following required amount of lever presses doubled after every reward earned. Session was considered terminated when animals failed to lever press for three minutes, thus defining the session duration. PRx2 tasks were intercalated by RR20 sessions to prevent operant responding extinction. Mice underwent one PRx2 task without injections for basal performances. Then 4 PRx2 tasks were carried out, during which each mouse received CNO (2 mg/kg or 0.5 mg/kg; IP) or saline IP injection. Order of Saline/CNO administration was counterbalanced within groups. The 2 sessions of PRx2 under CNO and under saline were averaged for each animal (two PRx2 per treatment condition; within subject repeated-measures). The same procedure was used for experiments with JHU37160.

**Licking microstructures.** A homemade setup was used to record licking microstructures[42,65]. The test was run in polypropylene cages equipped with a feeding bottle and a metal grid floor connected to a computer with a dedicated software. The animal's contact between the grid floor and the metallic sipping tip of the bottle allowed for the precise detection of licks. Mice were habituated to the testing chamber with bottles filled with water for 2 days through 30 min sessions. On the test day, the bottle was filled with 10% condensed sweet milk diluted in tap water and mice were injected with CNO (2 mg/kg) or saline solution 30 min before the beginning of the 30-min testing session (within subject repeated-measures). The system allowed for the detection of the total number of licks, total number of bursts and number of licks per burst. Bursts were defined as sequences of high frequency licks, in which the interlick interval was less than 250 ms. Relation between number of licks per burst and hedonic reactivity have been demonstrated in[42,66,67].

**Motor coordination in rotarod.** A rotarod apparatus (Bioseb) was used to assess motor coordination under CNO (2 mg/kg) and saline (within subject repeated-measures). Mice were placed on the rod rotating at 4 rpm. During the test, the rotation of the rod increased from 4 rpm to 40 rpm in 300 sec. The latency to fall was recorded.

**Measure of 48-hour ad libitum food intake.** Single-housed mice were food deprived for 2 hours and were injected with CNO (2 mg/kg; IP) 30mn before the onset of the dark phase. Chow pellets (A04, SAFE) were placed in a dish into the cage. Food dishes were weighed after 1; 2; 4; 12; 24 and 48 hours. The same procedure was performed with animal receiving simultaneous CNO (2 mg/kg; IP; within subject repeated-measures) and leptin infusions (5 mg/kg; IP) 3 hours before dark phase, or CNO (2 mg/kg; IP) and ghrelin infusions (3 mg/kg; IP) at the beginning of the light phase (9.30 AM). In this latter condition, chow pellets were placed in a dish into the cage 30 minutes after injections (10.00 AM).

**Fasting Refeeding procedure.** Single-housed mice previously exposed to high-fat diet (Research Diet ref: D12451), were food deprived for 12 hours, then injected with CNO (2 mg/kg or 0.5 mg/kg; IP) 30mn before the onset of the dark phase. Following high-fat diet access, food was weighed at 1; 2; 4; 12; 24 and 48 hours.

**Pavlovian conditioning for the measure of food intake coupled with fiber photometry recordings.** Food-deprived mice were placed in the operant chamber for 20 min sessions, during which a drop of sweetened condensed milk was delivered 3 seconds after the emission of a sound (65 db, 3000 Hz, 7 s) and the illumination of a diode lasting 10 seconds (ITI: 45 s after reward consumption). Animal were first trained during 10 sessions. For the measure of food intake, during the last two sessions, animals received i.p. saline or CNO (2 mg/kg) injections in a counterbalanced manner, 30 minutes before the trials. Licking patterns were analyzed as a proxy for reward consumption. The same procedure was used for the recording of calcium transients in animals expressing GCaMP8m in either D1- or D2-expressing neurons of the NAc. Implanted animals were connected to A RWD (R820 Fiber Photometry) fiber photometry setup. A 470 nm LED light stimulation coupled to a 405 nm isobestic stimulation was delivered through a fiber-optic patch cord (O.D. 1.25 mm, Ceramic Ferrule, 400 μm core diameter, 0.5 NA; R-FC-L-N5-400-L1, RWD). Intensities were set to reach a 100 μW power for 470 nm LED and 40 μW for 405 nm isobestic LED. Activity was recorded at 40 Hz. Mice were subjected to 7 sessions of 15 minutes. For recording of dopamine transients, the same behavioral paradigm and recording settings were used. The effect of CNO was assessed after extensive pavlovian training (10 sessions) over 2 sessions, intercalated with one session with saline injections (3 total with one session before and one after the CNO injections). The z-score around each behavioral event was then calculated using a baseline of 1.5 sec (from −2sec to −0.5 sec before the event) as $\frac{\Delta F/F - (means\ of\ values\ in\ baseline\ period)}{standard\ deviation\ of\ values\ in\ baseline\ period}$. Area Under the Curve (AUC) were calculated in MATLAB using the trapz function.

**In vivo fiber photometry recording of calcium transients during voluntary wheel running or free-feeding.** Animals expressing GCaMP8m in either D1- or D2-neurons of the NAc (see above) were singled-housed with *ad libitum* access to a running wheel and *ad libitum* food during 2 weeks. Two days before testing, either the running wheel was removed or animals were food restricted. The behavioral apparatus consisted in a four-chamber maze composed of two waiting areas each leading to distinct compartments containing either palatable grain-based food pellets (Dustless Precision Pellets, Bio-Serv) or a running wheel. A session consisted in recording baseline activity in the waiting area for 60 sec, then the door was opened giving access to either the running wheel or 6 food pellets distributed every 30 sec. Animals were recorded during 3 min. 2-3 sessions per day were performed (for either the running wheel or food consumption) during 6 consecutive days. Animals' behavior was video-recorded synchronized with the fiber photometry recording (TTL), and a post-analysis of behavioral epochs identified through the BORIS Software, allowing the alignment with fiber photometry signal and a peri-event analysis.

Fiber photometry data were acquired with Doric Neuroscience Studio software through a fiber photometry console controlling excitation lights, demodulated fluorescence signals and received timestamps of behavioral events (Doric Lenses). Fiber photometry uses the same fiber to both excite and record from animal via optical fiber patch cord (RWD Optic Fiber 0,5 NA, 400 μm flat tip, L:1 m). Blue excitation light (460 nm Doric Lenses LED) and isosbestic light (405 nm Doric Lenses LED) passed through a filter cube (iFMC4; Doric Lenses). Light intensity at the tip of the fiber was measured before every recording session and kept at 40 μW for isosbestic light and at 100 μW for blue excitation light. During recording, both light stimulations were alternatively triggered at 10hz. Then fluorescence signals were collected through a Fluorescence Detector Amplifier (DC x1,Doric Lenses). Signals were sampled at 12048.19 Hz.

Fiber photometry signals were analyzed using custom MATLAB (MathWorks) scripts. Signals were demodulated according to the 460 nm and the 405 nm light emission and resampled at 10 Hz. The

isosbestic signal (405 nm signal) was scaled using a least-squares linear regression to best fit the 460 nm signal. This scaled isosbestic signal was subtracted from the calcium-sensitive signal at 460 nm to compute a movement corrected, and bleaching-corrected change in fluorescence (ΔF), then the ΔF/F was calculated by dividing the change in fluorescence by the scaled isosbestic signal. To detect calcium transients, the ΔF/F signal was bandpass filtered (4th-order Butterworth filter: lower cutoff frequency 0.1 Hz, higher cutoff frequency of 2 Hz), and calcium activity peaks were detected when the signal exceeded a threshold defined by the median value + 2 median absolute deviation (MAD) with a prominence of at least one MAD and a distance between peaks of at least 0.1 s. For each behavioral event, the photometry signals were z-scored using a pre-event baseline ranging from −40 s to −30 s for 'feeding room entrance', 'feeding period', 'running room entrance', 'running period' events, or from −8 s to −5 s for 'feeding', 'stop running', and 'locked wheel' events. Resulting photometry signals were then analyzed to first determine significant event-related transients around behavioral events in either D1- or D2-neurons. For each event, using the compilation of peri-event z-scored photometry signals, a bootstrapping procedure was used to calculate confidence intervals (95% CI, 2000 random resampling with replacement). A 95% CI was calculated for each time point using the 2.5 and 97.5 percentiles of the bootstrap distribution and was then expanded by a factor of $\sqrt{(n/(n-1))}$ to account for narrowness bias[68,69]. Significant transients were defined during periods for which the 95% CI does not contain 0 (corresponding to the pre-event baseline) for at least 1 s. Second, we compared photometry signals recorded from D1-neurons and D2-neurons around each behavioral event using permutation tests (2000 random partitions)[70]. Signals were defined as significantly different if the exact permutation-based p-value stays below the significance threshold set at 0.05 for at least 1 s. In addition, the area under curve (AUC), the average z-score, the peak z-score, and the average transient rate were calculated respectively as the sum of the z-scored photometry signal, the averaged value of the z-scored signal, the maximum value of the z-scored signal, and the average frequency of detected calcium transients. For each behavioral event, these values are calculated during a pre and a post periods as indicated in the figures.

**Effect of acute manipulation of neurons on wheel-running/free-feeding.** The setup consisted of a magnetic fast-track running wheel mounted onto a mouse igloo shelter (BioServ) and connected to a magnet odometer. Single-housed mice had free access to regular chow and the running wheel for 2 days for habituation. On the test day, mice were fasted for 12 h then placed in the cage with the running wheel and *ad libitum* chow placed inside the cage for 2 h for baseline consumption and wheel activity. CNO (2 mg/kg) or saline (within-subject repeated measures) was then injected and food intake and wheel activity were recorded every hour for 3 h. For each condition (pre- or post-infusion), data are expressed as the difference between CNO and saline condition, divided by the value under saline condition. Animals with at least one outlier ratio value (Grubb's test) were excluded from analysis.

**Effect of chronic manipulation of D1- and D2-neurons on wheel-running/free-feeding**

Single-housed mice were exposed to a running wheel and standard diet for 1 week prior to the beginning of the test. 24-hour wheel running activity, food consumption and weight variation were recorded and defined as a baseline. On the first day of the test animals' food was switched to HFD and CNO (2 mg/kg) was administered every day before the onset of the dark period for 23 days, during which variation of percentage of the aforementioned parameters were evaluated. Measurements were pursued for 10 days with no CNO administration. Subsequently, mice were subjected to progressive food restriction

over 7 days (decrease of 0.5 g per day, starting at 3 g) with a daily CNO infusion and free access to the running wheel.

### Effect of concomitant manipulation of D1- and D2-neurons in the ABA model
Mice were individually housed and given free access to a running wheel, food, and water for 7 days (day −7 to day 0). On day 0, 3 h into the dark phase, food was removed, and mice were placed on scheduled feeding of 3 h per day starting at dark onset. CNO was injected 30 minutes before addition of the food.

### MRI for body composition
The last day of chronic chemogenetic manipulation experiments, whole body composition of was measured (basal condition, after CNO treatment period and after CNO withdrawal period) using EchoMRI, minispec LF90II (Bruker). It acquires and analyzes TD-NMR signals from all protons in the entire sample volume and can provide 3 components of interest: Fat, Free Body Fluid and Lean Tissue values. Total mass and fat mass variation between each period were calculated.

### Glucose tolerance test
Single-housed mice were food deprived 5 hours prior to the test at 09:00. Glycemia was measured using single-use test strips and glycemia meter (ACCU-CHEK Performa, Roche). The tip of the tail of the animal was cut for the first measurement, in order to obtain 1-2 drops of blood. For the following blood tests, the tail was massaged from the base to the tip so that a small volume of blood comes out. The first drop was wiped away and the second drop was used to measure glycemia. Mice were injected with CNO (2 mg/kg; IP) or saline 20 minutes prior to the beginning of the test. Glucose (2 g/kg; IP) was injected to the mice at t=0mn. Nine blood samples were collected throughout the test (−30mn before CNO/Saline injection; 0mn before glucose injection; +15mn; +30mn; +45mn; +60mn; +90mn; +120mn; +180mn) and mice were left quiet in a closed soundproof room in between samples.

### Plasma dosage of ghrelin and leptin
At the end of the progressive food-restriction procedure, CNO was injected 90 minutes before blood sample collection in EDTA precoated tubes through cardiac puncture. Plasma was obtained by centrifugation (4500 rpm, 5 min at 4 °C) and then stored at −80 °C until further analysis. Plasma concentrations of acylated ghrelin and leptin were determined by commercial ELISA kits according to the manufacturers' instructions (Spi Bio #A05117.96 and Millipore #EZML-82K, respectively).

### Immunohistochemistry
Mice were deeply anesthetized using a lethal IP dose of sodic pentobarbital (Exagon, 300 mg/kg) together with lidocaine (Lurocaine 30 m g/kg), and transcardially perfused with an ice cold 4% paraformaldehyde solution in PBS 1X. Brains were removed and immersed in 4% PFA overnight at 4 °C, then transferred in cryoprotectant solution (30% glycerol; 30% ethylene glycol in PBS-1X pH 7.4), and stored at −20 °C until further use. Brains were rinsed in PBS 1X for 30 min before slicing. 40 μm coronal sections were collected using a vibratome (Leica) into cryoprotectant solution and stored at −20 °C until further use. Brain slices were mounted on slides and coverslipped with VectaShield containing DAPI mounting medium (Vector Labs). mCherry expression was visualized with a Hamamatsu NANOZOOMER 2.0HT microscope (Bordeaux Imaging Center facilities). For mCherry signal amplification, brain sections were rinsed in PBS-1X, treated with blocking buffer (10% Fetal Bovine Serum in TBS: Tris 0.1 M, NaCl 0.9%, pH 7.4) for 2 h at room temperature (RT) and incubated overnight at 4 °C with primary antibody against mCherry (rabbit; 1:1000). For GFP signal amplification primary antibody against EGFP (mouse; 1:1000). Sections were then incubated with the secondary antibodies (donkey

anti-rabbit; 1:1000 and goat anti-mouse; 1:1000) conjugated to a fluorophore for 2 hours RT and mounted on slides and coverslipped with VectaShield containing DAPI mounting medium and the signal was visualized with a Hamamatsu NANOZOOMER 2.0HT microscope. To evaluate viral expression in cholinergic interneurons, the procedure was repeated with a primary antibody against ChAT (Choline acetyl transferase; goat; 1:500) and a secondary antibody (donkey anti-goat; 1:1000).

### Statistical analysis
Data are expressed as the mean $\pm$ SEM and statistical analyses were conducted using Prism software with a threshold for statistically significant differences set at $p \leq 0.05$. Each figure represents behavioral tests conducted on different male animal cohorts. The results were analyzed using adequate statistical and post hoc tests with repeated measurement when appropriate (see Supplementary Table 2 for details). Outlier values were removed with Grubbs' method when appropriate (see Supplementary Table 2 for details).

### Reporting summary
Further information on research design is available in the Nature Portfolio Reporting Summary linked to this article.

## Data availability
The data that support the findings of this study are available within this article and its Supplementary Materials and are available from the corresponding authors upon request. Source data are provided with this paper.

## Code availability
Custom codes used for this study are available here: https://github.com/Trifilieff-Lab/Walle_2024_NatCommun with https://doi.org/10.5281/zenodo.10669866.

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

## Acknowledgements

We thank the Bordeaux Imaging Center funded by ANR-10-INBS-04 for imaging, Motac Neuroscience for lending the rotarod apparatus, Céline Ducroix-Crépy and Jean-Christophe Helbling (INRAE UMR 1286) for genotyping and the staff from the animal facility of INRAE UMR 1286 for animal care. This study was supported by INRAE and University of Bordeaux; Idex Bordeaux "chaire d'installation" (ANR-10-IDEX-03-02) (PT), University of Bordeaux's IdEx "Investments for the future" program/GPR BRAIN_2030 (PT), NARSAD Young Investigator Grants from the Brain and Behavior Foundation (PT), ANR "SynLip" (ANR-16-CE16-0022) (PT), ANR "FrontoFat" (ANR-20-CE14-0020) (PT), ANR "StriaPOM" (ANR-21-CE14-0018) (PT), Institut de Recherche en Santé publique (IReSP) Aviesan APP-addiction 2019 (PT), Labex "BRAIN" (PT and RW), Region Nouvelle Aquitaine 2014-1R30301-00003023 (PT); Fondation pour la Recherche Médicale (FRM "Environnement et Santé") (GF and PT), ANR "HA-CTion" (ANR-19-HBPR-0002) (GF and PT); PRESTIGE-2017-2-0031 (AC), "Fondation pour la Recherche Médicale" SPF201809007095 (AC); Canadian Institute for Health Research (201309OG-312343-PT; 201803PJT-399980-PT) (BG); FD and RW are recipients of PhD fellowships from the French Ministry of Research. AP is a recipient of a PhD fellowship from the "Ecole Universitaire de Recherche" (EUR Neuro, Bordeaux Neurocampus); AKE is Research Director of FRS-FNRS and WELBIO investigator. AKE was supported by grants from from FRS-FNRS (#23587797, #33659288, #33659296), Fondation Simone et Pierre Clerdent (Prize 2018) and Fondation ULB.

## Author contributions

R.W., G.F., F.D., F.G., and P.T. designed research, R.W., A.P., G.R.F., E.M., A.C., M.F.A., M.P., R.O., A.O., V.D.P., and F.D. performed research, R.W., F.D., F.G., and P.T. supervised research, R.W., A.P., G.R.F., C.V., E.M., L.H., A.C., M.F.A., M.P., F.D., F.G., and P.T. analyzed data, R.A.A., A.K.E., B.G., and F.C. provided expertise, reagents and material, R.W., F.C., G.F., F.D., F.G., and P.T. wrote the manuscript. All authors edited and approved the manuscript.

## Competing interests

The authors declare no competing interests.
