## [Peer Review File · Nature Communications]

Nucleus accumbens D1- and D2-expressing neurons control the balance between feeding and activity-mediated energy expenditureREVIEWER COMMENTS

Reviewer #1 (Remarks to the Author):

This very interesting work by Walle et al, shows that manipulation of NAc core D1- and D2-neurons alters food consumption, activity and weight of the animals. In particular, authors show that chemogenetic manipulation of these neurons exerts opposite effects in food consumption, with D2-neurons inhibition decreasing milk and chow consumption, and increasing activity; D2 activation leading to the opposite effect. Conversely, D1- neurons inhibition increased food intake and decreased activity; D1-neurons activation mirrored this. Moreover, authors provide fiber photometry recordings during activity and during feeding, and MRI measures of fat and lean. Authors also simultaneously manipulate both populations to test their hypothesis that chronic activation of D1-neurons + inhibition of D2-neurons leads to activity-related energy expenditure with weight/fat loss (and vice versa).

The message of the manuscript is important, and the experimental design is adequate. However, I have some concerns that require special attention. Please see my comments below, I believe that addressing these points can clarify and improve the message of the manuscript.

Methods

- Provide More details on the method to measure fat, lean
- Authors do not mention if they removed outliers or not, and if so (which appears to be the case because the number of animals is not the same in all experiments), what was the criteria, and provide details in the statistical table.
- Sex of mice?
- Were control wt animals also treated with CNO?
- Provide more details overall regarding the experimental design. It is not always clear which animals were used for what.
- How did you control for feeding epochs in the photometry recordings to then align to neuronal activity? This is not mentioned in methods.

Results

- I think that the inclusion of VTA data could be better supported by the authors, this looks disconnected from the rest of the data.
- Did authors perform other tests to evaluate the impact of the chemogenetic manipulation in motor behavior and locomotion?
- Do authors have an explanation for why the licks/burst is very different from D2- and D1 animals (S1D, S1K)?
- It is not clear if the PR was done immediately the day after the RR20?
- How was the PR data calculated? Average of the 4 PR sessions or is the result presented of one of the sessions? Is the behavior stable over time?
- The average of lever presses and breakpoint of Fig1H and S1H is very different. What is the possible explanation for this?
- Use of D2-cre line has some caveats as D2R is also expressed in 80% of cholinergic interneurons. Authors should discuss this, and change the text accordingly by using D1-neurons and D2-neurons to be more accurate. This is especially important considering data showing that cholinergic neurons may also control body weight and metabolism. Authors should also discuss this.

- Line 239: was the performance of the animals similar over time? How long did it take for animals to pass to the other ratio?
- Considering the half-time of CNO, and respective modulation of DREADDs, what is the possible explanation for the effect in food consumption only 48h after CNO administration (eg Fig2C)?
- Viral transfection extension should be provided for all animals in supplementary material.
- Fiber photometry data of Fig. 3 and S3 needs to be revised and presentation improved, some text is not readable so I could not understand what was depicted. There are other methods of analysis of this type of data that could also be included (auc, average activity during a specific period, amongst other).
- Line 470-471: what was the statistical analysis performed to compare D1- and D2-MSNs activity? permutation test? This is not clear.
- Because photometry recordings were done in different animals for D1 and D2 neurons, do authors have recordings in a "control" behavior not related to food and running?
- I think authors could put all the photometry data in main figure.
- Did authors perform correlation of activity with performance? i.e. activity with wheel running velocity? Or with feeding time?
- Did authors control for differences in locomotion or motor behavior?
- What is the explanation for D2Gi animals also presenting higher fat mass? Is this not significant in comparison to WT?
- Did authors test the effects of D1Gi-D2GI and D1Gq-D2Gq manipulations? What would authors expect?
- Line 622- 623 – several studies do not report any effect of MSN manipulation in food intake. Any explanation on this?
- Title: Since cholinergic interneurons also express D2R, the title should not include MSN, maybe D1- and D2-neurons; also I would remove "network"
- Did authors perform correlations between neuronal activity from photometry data and running/feeding? Or correlations between the fat/lean content and weight variation and/or running and/or food consumption?
- Considering some previous studies in Nac shell D1 and D2 neurons, is possible that the effect observed is due to changes on peripheral signals controlling feeding and energy balance. Did authors measure any molecules in the periphery of chronically manipulated animals? This could strengthen the message of the work.
- Previous studies have shown that NAc or dorsal striatum D2 neurons ablation caused overconsumption of calories, and further suggesting that D2 neurons send appetite-suppressing signals (Cell reports 2023). In fact, there anatomical and functional connections with circuits associated with appetite suppression for D2 neurons. How to conciliate these results with this work?
- Authors strongly emphasize the importance of these findings in the context of eating disorders. Considering that in the last few years there has been some publications related to these findings, I

believe that the manuscript would stand out by including similar experiments in a model of ED for example, as a proof of concept. For example, does manipulation of the MSNs reverts the phenotype of any of the available ED models? Or authors could perform fiber recordings in lean/obese or ED model to show de-regulation of MSNs? This would highlight the clinical relevance of these findings.

- What is the proposed working model? What are the possible output targets of these manipulations?

- From the available data it is clear that the NAc neurons are important for feeding and energy balance, and that the same neuronal subtype of different subregions may play different roles. This could be better highlighted in the discussion.

- Considering your working model, maybe it would be interesting to discuss these findings in the light of the known connections with the VTA and LH (or other appetite/energy balance brain regions)?

Minor comments

- I suggest separating the description of the different types of data in the first paragraph of results to make the reading easier.

- I think that a summary table or a schematics representation of the findings would be very interesting to have, even if in sup. material.

- Legend of colors in graphs of Fig2A and E (it only appears in figures below)

- Throughout the text, authors should refer to the supplementary figures with more detail: eg. Line 412 - Sup Fig1J, instead of SF1.

- Try to normalize axis whenever possible to allow better comparison between D1 and D2 effects: eg. Lever presses, breakpoint etc

- Line 223: through?

- Line 406: change amotivational

- Typo in S2D: salin instead of saline; refer to more details about this in methods.

- Authors should refer if the researchers were blind or not to the procedures.

- Line 452 rephrase

- Fig. 4D : using the nomenclature "after CNO" it is not clear as it looks like authors are referring to the period when CNO was removed. I would just put CNO period

- Overall, I suggest to improve figures representation (this is a personal comment, not a scientific one); especially photometry data.

- Scale bars missing in figures (eg 3D, 5A...); 3D needs to be larger, the figure of the box is not very clear, maybe a legend would help

- Fig4a missing time

- Include references of food, palatable pellets

- Line 492: refer that these measures were obtained by MRI.

- Maybe change Withdrawal with after CNO suspension or CNO removal?

- For all graphs include the individual dispersion points to highlight variability.

- Provide better figures for 5A; mice scheme can be improved by adding the virus injected; figures missing scale bars and small, low quality, the whole NAc should be presented

- Line 547-548: I don't think there are enough experiments to support this affirmation as it stands.

- I would uniform axis of the same parameters to be easier to compare between them, whenever possible.

Ana João Rodrigues

Reviewer #2 (Remarks to the Author):

Review of NCOMMS-23-18491

Walle et al. are investigating how D1 and D2 accumbal core MSNs regulate behavior in an operant task. This is an important area of study as the precise push-pull of these cell populations in driving motivated behavior is not fully understood. While the authors show some exciting data suggesting that D1 and D2 MSNs in the NAc drive operant performance for food rewards, along with bidirectional modulation of feeding and wheel running, the study suffers from lack of hypothesis-driven rationale, overinterpretation of experimental results, and at times is missing control groups from experiments. Throughout the paper, experiments are presented and results stated, but often in the absence of specific hypotheses to inform the experimental plan, or without conclusions formed about the experimental results. Additionally, while obesity and anorexia inarguably affect similar mesolimbic circuits, the premise of the paper, that obesity and AN are opposite ends of the spectrum is reductive. For these reasons it is this reviewer's opinion that at this time this paper is not suitable for publication. Concerns are outlined below:

- Vague language is used in the abstract – for instance in lines 49-51 “effects” are referenced.
- Line 106 – ref 34, they say chronic disruption of d1s leads to obesity. The reference in question showed that chronic inhibition of d1s is somewhat protective from DIO
- For Figure 1, It looks like for the D1s, CNO might be the effect? Data from experiments using control GFP viruses should be presented and analyzed alongside the dREADD groups
- For data presented in Figure 2, the authors looked at how inhibition and activation of NAc d1s and d2s affected ad lib chow intake, and saw that inhibiting d2s decreased chow consumption up to 48 hours, while activating d2s only enhanced chow consumption 2 days later. Why do they think it would last two days but not have an effect initially? For instance, are they activating lasting plasticity mechanisms that are in turn driving hunger? This data should be further explained or removed from the narrative.
- Is the same dREADD cohort of mice used for experiments shown in figures 1-4? If so, correlative analyses could be used to explore whether there are relationships between variables, for instance whether performance on operant assays correlates with food consumption, etc.
- They recorded bulk calcium activity via fiber optics in the nac during feeding or wheel running. The analysis as presented in fig 3e is referred to as revealing that during consumption d2 activity “Remains significantly higher” than d1s (line 470). But only perievent histograms are shown, but not an averaged data set on which statistics were run.
- Fig 3e-g have red, blue, and black horizontal lines on the graphs but it is not explained what these are.
- In the manuscript, pubmed IDs instead of citations
- Zhu 2016 (frontiers) already showed that chemogenetic modulation of d1 and d2 nac neurons bidirectionally drives wheel running
- Weight graphs should be plotted over time. So should distance travelled; right now just “food consumption” and “wheel running variation” is plotted (right now they are just listed in a supplemental table).
- Chronic dREADD manipulation is not advised for long-term experiments as they lose efficacy via receptor desensitization with frequent dosing over weeks (Roth 2016, PMID: 26889809).

Reviewers' Comments:

Reviewer #1: This very interesting work by Walle et al, shows that manipulation of NAc core D1- and D2-neurons alters food consumption, activity and weight of the animals. In particular, authors show that chemogenetic manipulation of these neurons exerts opposite effects in food consumption, with D2-neurons inhibition decreasing milk and chow consumption, and increasing activity; D2 activation leading to the opposite effect. Conversely, D1- neurons inhibition increased food intake and decreased activity; D1-neurons activation mirrored this. Moreover, authors provide fiber photometry recordings during activity and during feeding, and MRI measures of fat and lean. Authors also simultaneously manipulate both populations to test their hypothesis that chronic activation of D1-neurons + inhibition of D2-neurons leads to activity-related energy expenditure with weight/fat loss (and vice versa).

The message of the manuscript is important, and the experimental design is adequate. However, I have some concerns that require special attention. Please see my comments below, I believe that addressing these points can clarify and improve the message of the manuscript.

We are grateful to the Reviewer for her appreciation of our work and for the very constructive comments that we have tried to address by providing several additional experiments and editing of the manuscript. We would like to emphasize that several very relevant aspects raised by the Reviewer would definitely deserve more development in the discussion. However, due to space limitation (5000 words for the main text) and the significant amount of new data we provide, we could not be as exhaustive as we would have liked. We hope the Reviewer will understand this constraint.

Methods:

1- Provide More details on the method to measure fat, lean

We added additional details in the method part (line 728-134).

2- Authors do not mention if they removed outliers or not, and if so (which appears to be the case because the number of animals is not the same in all experiments), what was the criteria, and provide details in the statistical table.

Outliers were indeed removed based on statistical analyses (Grubbs' test), or following histological analyses (off-target expression mostly). This was indicated in Table 2 of the original version of the manuscript and we now ensured that this was clearly stated when applicable.

3- Sex of mice?

In the "experimental models and subject details" section of the original version of the manuscript, we indicated that male mice were used. However, according to the guidelines of *Nature Communications* it is now clearly stated in the abstract of the revised version of the manuscript.

4- Were control wt animals also treated with CNO?

We apologize for not making this important point clearer. In fact, for chronic chemogenetic manipulations where WT littermate mice were used, all mice were treated with CNO to rule out potential CNO effect that are not dependent on DREADD expression. We now clearly state it in the method section (line 538) and the results of the revised manuscript (line 259).

- Provide more details overall regarding the experimental design. It is not always clear which animals were used for what.

We are sorry for not making this point clearer. All experiments were done with different cohorts, except for analysis of licking microstructures which were performed in the same animals that underwent operant conditioning (Figure 1 and SF1 C,J). In addition, animals subjected to the

progressive food deprivation (Figure 6 A-C) were previously subjected to chronic chemogenetic manipulation (Figure 5). This is now clarified in the methods section (line 721-722).

- How did you control for feeding epochs in the photometry recordings to then align to neuronal activity? This is not mentioned in methods.

We apologize for not mentioning that in the original version of the manuscript. Animals were recorded with a video synchronized with fiber photometry recording. Analysis of behavioral epochs was performed *a posteriori* through the BORIS Software, allowing further alignment with fiber photometry signals and peri-event analysis. Details are now provided in the methods section (line 659-662).

Results:

1- I think that the inclusion of VTA data could be better supported by the authors, this looks disconnected from the rest of the data.

The Reviewer is right that, as it was, these data were a bit disconnected from the rest of the findings. We initially performed these recordings to ensure the validity and efficacy of our chemogenetic manipulations from a “circuit prospective”, showing that activation of NAc D1- and D2-expressing neurons increased and decreased the firing of VTA dopaminergic neurons, respectively, accordingly to the expected projections of NAc MSN subpopulations.

However, the Reviewer’s comment encouraged us to further explore the functional consequences of these chemogenetic manipulations on mesoaccumbens dopamine transmission, since the NAc-to-VTA pathway has been shown to be involved in feeding behavior (Bond et al., 2020; PMID 32354856). Using the dopamine sensor dLight 1.3 coupled with fiber photometry, we assessed how chemogenetic activation of either D1- or D2-expressing neurons of the NAc would impact mesoaccumbens dopamine dynamics in a Pavlovian conditioning task in which a tone cue predicts the occurrence of a palatable food reward. As expected, after repeated associative learning sessions, dopamine responses were strongly aligned with the occurrence of the predictive cues, with a response remaining during reward delivery/consumption. After extensive training, we chemogenetically activated either D1- or D2-expressing neurons of the NAc and assessed the impact on dopamine dynamics.

We found that chemogenetic manipulations of NAc D1- or D2-expressing neurons had no effect on mesoaccumbens dopamine dynamics during the cue presentation. Chemogenetic potentiation of activity of D1-expressing neurons had no effect on dopamine transmission during reward consumption. However, chemogenetic activation of D2-expressing neurons increased dopamine transmission during consumption of the reward. Thus this approach allowed us to assess whether changes in feeding were directly related to consumption and/or to alterations of the association between food-predicting cues and/or changes in the rewarding value of the food, both being mediated by dopamine signals. Altogether, these findings support that changes in feeding in response to chemogenetic manipulations of D1- or D2-expressing neurons are unlikely to be related to an alteration of associative processes since dopamine response during a reward-predicting cue is not altered. However, the enhanced signal during activation of D2-expressing neurons suggest a potential effect on reward value that could participate to increased feeding, even though we did not see any effect of the manipulation of D1-expressing neurons, suggesting that another mechanism might be at play. From a circuit prospective, this experiment also suggests that the effects of chemogenetic manipulations on feeding are not mainly mediated by changes in dopamine transmission (see below point #25). Moreover, to further strengthen these findings, we performed an additional experiments in which we recorded the bulk activity (GcAMP) of D1- or D2-expressing neurons using the same pavlovian conditioning paradigm. We show that, while the 2 subpopulations are comparably activated during CS presentation, activity of D2-expressing neurons rises during the first lick for consumption of the reward, but not for D1-expressing neurons – that tends to be inhibited. These data further support the opposite roles of these 2 subpopulations in feeding.

All these new findings are now presented in Figure. The electrophysiological data are now presented in supplementary figure 2 as an illustration of the functional effect of chemogenetic manipulation of NAc D1- or D2-expressing neurons.

2- Did authors perform other tests to evaluate the impact of the chemogenetic manipulation in motor behavior and locomotion?

We initially provided data in supplementary figure 1E and 1L showing that chemogenetic manipulations of D1- or D2-expressing neurons had no effect on motor coordination in a rotarod. Nonetheless, to address the reviewer's concern, we performed an additional experiment showing that, in an open field, on the one hand chemogenetic activation and inhibition of D2-expressing neurons decreased and increased spontaneous locomotion, respectively. On the other hand, chemogenetic activation of D1-expressing neurons potentiated locomotion, while inhibition had no significant effect (Supplementary figure 1E and I of the revised manuscript).

These data mainly parallel our findings regarding voluntary wheel-running and corroborate the correlation between distance moved in an open field and distance run on wheel in mice (Careau et al., 2012; PMID 22573112), that has been linked to the activity of dopamine transmission (Fernandes et al., 2015; PMID 26341832).

3- Do authors have an explanation for why the licks/burst is very different from D2- and D1 animals (S1D, S1K)?

We do not have much of an explanation but believe this is rather related to a cohort effect. We routinely use this parameter as a proxy for hedonic reactivity under various genetic, nutritional or pharmacological manipulations and found that the baseline value can strongly differ between cohorts. For example, in a study exploring the effect of fatty acid manipulations in a C57Bl6J background (Ducrocq et al., 2020; PMID 32142670) the average value in n-3 PUFA deficient animals and their controls was between 10-15 licks/burst (Ducrocq et al., 2019; PMID 30597761). In another study in which we compared high-fat diet exposed animals with control diet in C57Bl6J, the average value was around 25 licks/burst, with no difference between groups. Herein, D1-cre mice displayed an average value of approximately 20-25 licks/bursts, while it was around 15 for the D2-cre expressing the DREADD Gq but around 5 for the group expressing the DREADD Gi. Though we agree the difference between Gi and Gq-expressing D2-cre mice is quite surprising as the 2 groups were littermates and run in parallel. One possibility is that the repeated activation of the DREADDs – as this cohort was previously ran in operant conditioning – might have triggered changes that subsequently impacted licking pattern. Nonetheless, we did not find any effect of CNO injection in a within-subject manner which comforts us that these manipulations had no effect on licking pattern *per se*.

4- It is not clear if the PR was done immediately the day after the RR20? We apologize for not making this clearer. Indeed, all the PR sessions were performed the day after an RR20 session. We now clarify this point in the methods (line 587-591).

5- How was the PR data calculated? Average of the 4 PR sessions or is the result presented of one of the sessions? Is the behavior stable over time?

We apologize for not making this point clearer and now better explain in the method section (line 591-593). After 2 PR sessions with saline injections (to habituate the animals to the injection), intercalated with RR20 sessions, 4 PR sessions – still intercalated with RR20 sessions – were ran, 2 with saline injections, 2 with CNO injections, in a counterbalanced manner. The graphs presented therefore correspond to an average of 2 sessions for the saline condition and 2 for the CNO condition. It is actually because performance can sometimes vary across sessions – and that animals were injected in a counterbalanced manner - that we averaged 2 sessions for each type of injection (saline vs. CNO).

6- The average of lever presses and breakpoint of Fig1H and S1H is very different. What is the possible explanation for this?

This is again a cohort effect that we often observe in our hands. Though, since for a given graph all animals were run in parallel, and analyses of chemogenetic manipulations performed in a within-subject manner, we believe this should not influence interpretation of the findings.

7- Use of D2-cre line has some caveats as D2R is also expressed in 80% of cholinergic interneurons. Authors should discuss this, and change the text accordingly by using D1-neurons and D2-neurons to be more accurate. This is especially important considering data showing that cholinergic neurons may also control body weight and metabolism. Authors should also discuss this.

The reviewer is absolutely right that the D2R is also expressed in a large majority of cholinergic interneurons. However, we (Ducrocq et al., 2020 PMID 32142670; this study) and others (Gallo et al., 2018 PMID29540712) have observed that, when using viral-mediated strategies – at least in the NAc -, very little recombination is found in cholinergic interneurons using the D2-cre line. Gallo et al. (2018) quantified that only 1 out of 10 ChAT-positive interneurons would express the viral transgene. Moreover, while they found an effect on motivational performance by overexpressing the D2R through viral-gene transfer in the D2-cre line, they did not observe any effect when using the ChAT-cre line, supporting that the effect was mostly mediated by D2-expressing MSNs. This is why we initially considered our effects to be mostly originating from D2-expressing MSNs.

Nonetheless, we agree with the reviewer that we cannot completely rule out a potential implication of cholinergic interneurons in our effects, in particular regarding food intake. We therefore changed the nomenclature throughout the manuscript and figures to D1-neurons and D2-neurons. In the results part we added a sentence stating: “*Of note, as previously described, we found very little viral-mediated expression of cre-dependent DREADD, if any, in ChAT-positive neurons (SF1A), supporting that cholinergic interneurons were mainly spared by DREADD manipulations*” (line 136-139).

However, increasing CINs activity reduces palatable food consumption while their inhibition had the opposite effect (Aitta-Aho et al., 2017 PMID 28497110), in contrary to our findings with manipulations of D2-neurons. This suggests that most of our effects under D2-neuron manipulations are likely to result from alteration of D2-MSNs' activity. We now discuss this aspect in the revised manuscript (line 440-446).

8- Line 239: was the performance of the animals similar over time? How long did it take for animals to pass to the other ratio?

We apologize for not making this clearer. For a given operant ratio animals usually reach stable performance within 3 sessions. On average, animals were subjected to a given ratio for 4-5 sessions before switching to a higher one. As written in the initial version of the manuscript, we ensured that animals had stable performance across 2 consecutive days, i.e. with less than 10% difference in the amount of lever pressing. This is now stated in the method section (line 579-581).

9- Considering the half-time of CNO, and respective modulation of DREADDs, what is the possible explanation for the effect in food consumption only 48h after CNO administration (eg Fig2C)?

The Reviewer is raising an important point. As we originally explained in the manuscript, we believe that, when in their housing cage, the orexigenic effect of chemogenetic activation of D2-expressing neurons is initially overshadowed by the strong decrease in motivation. In other words, the drive to retrieve the chow pellets (climb on the grid to collect the food) is blunted. It is only afterwards that the orexigenic effect takes over, leading to increased consumption and weight gain. This is supported by our findings showing immediate increase in consumption when using a sweet milk solution delivered through a dipper, which requires little effort (now Fig. 2B). We now tried to explain better this aspect and wrote: “*We interpret this latter finding*

as a rebound effect resulting from an initial overshadowing of the orexigenic effect of D2-neurons activation by the blunted motivational drive, decreasing the willingness to exert effort to retrieve the food pellets despite the increase in hunger drive. This is supported by the observation that in the case of easily-accessible liquid food (milk), food consumption immediately increases as animals mostly stay at the dipper to consume (Figure 2B and SF2A)” (line 168-173).

10- Viral transfection extension should be provided for all animals in supplementary material. At the request of the Reviewer, we now provide photos of viral transfection extension for all animals in suppl. Material. Few animals are missing (less than 10 for roughly 170 animals shown) due to problems during tissue processing, premature death, etc.

11- Fiber photometry data of Fig. 3 and S3 needs to be revised and presentation improved, some text is not readable so I could not understand what was depicted. There are other methods of analysis of this type of data that could also be included (auc, average activity during a specific period, amongst other).

We improved the presentation of these data, now in Figures 2 and 3 and provide other types of analysis, i.e. AUC, averaged z-score, peak z-score and transient rate.

12- Line 470-471: what was the statistical analysis performed to compare D1- and D2-MSNs activity? permutation test? This is not clear.

As written in the original version of the manuscript, we used the non-parametric permutation test to compare activity of D1- and D2-neurons. Signals were defined as significantly different if the exact permutation-based p-value stays below the significance threshold (0.05) for at least 1s. We also now provide AUC and average z-score for more global comparisons between D1- and D2-neuron activities.

13- Because photometry recordings were done in different animals for D1 and D2 neurons, do authors have recordings in a “control” behavior not related to food and running? The Reviewer’s point is fair. However, as depicted in Figure 3A-B of the revised manuscript, during the period in which the animals enter the rooms containing either the wheel or the food, we observe a strong rise in activity that does not differ between D1- and D2-expressing neurons. Moreover, while relative activity of D2-expressing neurons is higher than that of D1-expressing neurons during the feeding period, while it is the opposite during running – and that these measures were performed in the same animals -, it is unlikely that these relative differences between the 2 subpopulations originates from the fact that recordings were performed in different animals.

14- I think authors could put all the photometry data in main figure.

Following the Reviewer’s recommendation, all the fiber photometry data for feeding and running are now presented in Figure 2A-B, with additional analyses. Though we kept the heat maps and the data on “run stop” in supplementary figure 3E-G for space issues.

15- Did authors perform correlation of activity with performance? i.e. activity with wheel running velocity? Or with feeding time?

The Reviewer raises an interesting point. Unfortunately, with the home-made setups we used we do not have access to this kind of parameters.

16- Did authors control for differences in locomotion or motor behavior?

We are not completely confident to understand what the Reviewer is referring to. If the question relates to chemogenetic manipulations, as mentioned above, we originally showed absence of effect in a rotarod for motor coordination, and now provide data showing effects on spontaneous locomotion that corroborate our findings with running wheels. If the Reviewer refers to the calcium imaging data, as acknowledged in the previous point, we unfortunately do not have these kind of parameters.

17- What is the explanation for D2Gi animals also presenting higher fat mass? Is this not significant in comparison to WT?

We apologize for this mistake. The difference is indeed significant and now corrected on Figure 4D of the revised manuscript. Our interpretation is that these animals tend to eat more during the initial period of CNO treatment due to their initial increase in running. This is now mentioned in the results (line 268-270).

18- Did authors test the effects of D1Gi-D2GI and D1Gq-D2Gq manipulations? What would authors expect?

The Reviewer's point is interesting, however we did not test this. One could assume that D1Gi-D2Gi manipulation might resemble to some extent the effect of NAc lesion or pharmacological inhibition, that have been shown to decrease operant responding (see for instance Bezzina et al., 2008 PMID 18167622; Balleine and Killcross, 1994 PMID 7718151), increase feeding (Basso and Kelley, 1999 PMID 10357457; Maldonado-Irizarry et al., 1995 PMID 8590077; Reynolds and Berridge, 2001 PMID 11312311) – mainly linked to the activity of the shell subregion, although manipulation of the core of the NAc has been shown to mediate such effects as well (see for instance Herisson et al., 2016 PMID 27114001) - and decrease motivation for wheel running (Basso and Morrell, 2015 PMID 26052795). These findings mainly recapitulate the effects we observed through inhibition of D1-expressing neurons.

Regarding broad "activation", it has been shown for instance that glutamatergic agonist-mediated activation (Stratford and Kelley, 1998 PMID 9659985) or electrical stimulation (Houchonou et al., 2023 PMID 37919311) of the NAc reduces food intake and that electrical stimulation of the NAc enhances amphetamine-induced locomotor activity (Kokkinidis et al., 1989 PMID 2927262). Overall, these findings resemble the effect we obtained with activation of D1-expressing neurons.

Even though this remains to be tested with chemogenetic manipulations, this suggests that when both populations are modulated in the same "direction", alteration in D1-expressing neurons takes over changes in activity in D2-expressing neurons. This might be related to the strong lateral connection between dopaminergic subpopulations. However, these interpretations remain at this stage very speculative and we hope the reviewer will agree this goes beyond the topic of the current study that aims at addressing the effect of a change in the balance between D1- and D2-expressing neurons.

19- Line 622- 623 – several studies do not report any effect of MSN manipulation in food intake. Any explanation on this?

To the best of our knowledge, most studies focusing on the implication of MSN subpopulations in food-driven behaviors were based on optogenetic manipulations. Soares-Cunha et al. (2016; 2018) and Natsubori et al., (2017) did not seem to observe any effect on food intake, though, to the best of our knowledge, this was not directly assessed as short (seconds) optogenetic manipulations were mostly performed in a progressive ratio task – during lever pressing or reward-predicting cues - in which animals earn quite a limited amount of food reward and do not allow a direct measure of food consumption. Though, interestingly, in the 2022 paper by Soares-Cunha et al., authors manipulated the activity of D2-MSNs during reward delivery in the progressive ratio task. As the Reviewer knows, they found that their activation led to a decrease in breakpoint while their inhibition had the opposite effect, in accordance with our findings in operant conditioning using chemogenetics. However, to the best of our knowledge, the direct effect on food intake in a free-access procedure, with manipulation of D2-MSNs during reward consumption was not assessed. Yet, it is possible that, in operant procedures, the effect of MSN manipulations on the incentive component of behavior might overshadow the impact on hunger drive.

In line with this, using optogenetic manipulations in a free-access consummatory task (licking of a sweet solution), O'Connor et al. (2015) found that short stimulation of D1-MSNs of the NAc during consumption rapidly stop feeding. Similarly, Bond et al. (2020) found that longer stimulation (3 min) of D1-MSNs of the NAc decrease food pellet consumption in a free-access

procedure. These latter studies are in accordance with our findings of an anorexigenic role of D1-MSNs of the NAc.

20- Title: Since cholinergic interneurons also express D2R, the title should not include MSN, maybe D1- and D2-neurons; also I would remove “network”

As explained above, we agree with the Reviewer and changed the title accordingly. It now reads: “*Nucleus accumbens D1- and D2-expressing neurons control the balance between feeding and activity-mediated energy expenditure*”.

21- Did authors perform correlations between neuronal activity from photometry data and running/feeding? Or correlations between the fat/lean content and weight variation and/or running and/or food consumption?

As explained above, the setup we used for running and feeding coupled with fiber photometry experiments did not allow us a precise enough measurement of running or feeding bouts to be correlated with neuronal activity. As for correlations between body composition, weight variations, running and food consumption in our chronic manipulations, we did not find strong correlations, which is why we did not include these data in the manuscript. This might notably be due to the use of mice with variable ages (4-7 months) – and therefore with quite variable fat mass (see supplementary Figure 5C of the revised manuscript) -, due to the difficulty of generating large cohorts of double transgenic animals, necessary for these kind of experiments.

22- Considering some previous studies in Nac shell D1 and D2 neurons, is possible that the effect observed is due to changes on peripheral signals controlling feeding and energy balance. Did authors measure any molecules in the periphery of chronically manipulated animals? This could strengthen the message of the work.

We thank the reviewer for suggesting such an analysis. In the initial version of the manuscript we provided measures of blood glucose changes after acute manipulations of D1- or D2-expressing neurons (SF2) and found no effect. The Reviewer comment motivated us to look at other peripheral signals after chronic manipulation of striatal neurons. In line with the idea of strengthening the translational value of our findings as suggested by the Reviewer (see below), we analyzed plasma levels of leptin – an anorexigenic adipokine - and ghrelin – an orexigenic gut peptide - in a model of eating disorders, i.e. in animals previously subjected to a 5-days progressive food restriction, while having constant access to a running wheel, as a proxy for the imbalance between food intake and physical activity-mediated energy expenditure characteristic of anorexia. Leptin and ghrelin levels were analyzed 90 minutes after the last CNO injection. We found that the concomitant activation of D2-expressing neurons and inhibition of D1-expressing neurons – which tends to protect from weight loss – had no effect as compared to WT animals. However, leptin and ghrelin levels were significantly decreased and increased, respectively, in animals in which D1-expressing neurons were activated concomitantly to the inhibition of D2-expressing neurons and that displayed significant weight loss. These latter findings are consistent with what is described in patients suffering from Anorexia nervosa (Schorr and Miller, 2016 PMID 27811940). This suggests that changes in hormonal levels are a direct consequence of weight loss rather than being a consequence of D1- and D2-neuron manipulations that would influence the production of these hormones. Nonetheless, together with our pharmacological experiments of the effect of ghrelin and leptin injections on feeding that show that the effects of chemogenetic manipulations on food intake overcome the effects of hormones injections (now SF6A-B), this supports that manipulation of NAc subpopulations overcomes peripheral feeding signals. These data are now described in line 328-335.

23- Previous studies have shown that NAc or dorsal striatum D2 neurons ablation caused overconsumption of calories, and further suggesting that D2 neurons send appetite-

suppressing signals (Cell reports 2023). In fact, there anatomical and functional connections with circuits associated with appetite suppression for D2 neurons. How to conciliate these results with this work?

We assume the Reviewer is referring to the Sandoval-Rodriguez et al. (2023 PMID 36857179) study that we briefly mentioned in the discussion of the original version of the manuscript. Indeed, in this study, authors show that ablation of NAc D2 neurons renders animals less sensitive to appetite-suppressing signals, leading to increased food consumption. In this study ablation of either D1- or D2-expressing neurons, whether it is in the dorsal or ventral striatum had virtually no effect on spontaneous food intake. It is only in response to postingestive signals that the effects of ablation were unraveled. These findings are in fact very different – and even tend to be opposite to what we find. Of note, the effect they found regarding D1-MSN are also opposite to the studies of O'Connor et al. (2015) and Bond et al. (2020) (see above). We do not have any clear explanation regarding these discrepancies. One main difference – beyond that Sandoval-Rodriguez et al. used an ablation approach while the chemogenetic manipulation we used should be more transient – is that the extend of ablation in the NAc seems to concern roughly 40% of the targeted cells (60% intact). Considering the strong interplay between MSNs, it is possible that ablation of a limited number of neurons might lead to compensatory activity in remaining neurons as shown in many models of neuronal damage. However, we hope the Reviewer will agree that this is highly speculative and that, despite acknowledging the findings of Sandoval-Rodriguez et al. in the discussion as we did in the original manuscript, further studies will be required to disentangle the differences between their findings and those of others and we.

24- Authors strongly emphasize the importance of these findings in the context of eating disorders. Considering that in the last few years there has been some publications related to these findings, I believe that the manuscript would stand out by including similar experiments in a model of ED for example, as a proof of concept. For example, does manipulation of the MSNs reverts the phenotype of any of the available ED models? Or authors could perform fiber recordings in lean/obese or ED model to show de-regulation of MSNs? This would highlight the clinical relevance of these findings.

We really thank the Reviewer for acknowledging the importance and relevance of our findings in the context of eating disorders. We thought that the most striking effects of our findings was the ability of inducing weight loss when D1- and D2-expressing neurons were activated and inhibited, respectively as rodents usually adapt their food intake to energy expenditure. This is why, in the original version of the manuscript we presented data showing that chronic concomitant activation of D1-expressing neurons together with inhibition of D2-expressing neurons could precipitate weight loss in a food-restricted procedure in presence of a running wheel, as a proxy for the imbalance between food consumption and hyperactivity-induced energy expenditure characteristic of anorexia nervosa (Fig 6 of the revised manuscript). Nonetheless, this is not a “typical” rodent model of anorexia. Therefore, the Reviewer’s comment encouraged us to use a more accepted model and we therefore performed an “Activity-Based Anorexia” (ABA) experiment. The data are shown in Figure 6 of the revised version. In summary, we found that concomitant activation of D1-expressing neurons together with inhibition of D2-expressing neurons accelerates weight loss, while the opposite manipulation is “protective”. We hope this new set of highlights the clinical relevance of our findings as requested by the reviewer.

25- What is the proposed working model? What are the possible output targets of these manipulations?

This is a very fair comment from the Reviewer. Since, in this study, we did not explore the underlying circuits of our effects under NAc neuron manipulations, we can only stay very speculative which is why we only briefly discussed these aspects in the original version of the manuscript. Nonetheless the Reviewer’s request encouraged us to discuss these aspects. Below is the paragraph now added in the discussion (line 420-452): “*These latter findings raise the question of whether these two pathways are redundant to control food intake or modulate*

distinct – though complementary – dimensions of feeding behavior. It has been proposed that the NAc-VTA pathway modulate feeding through a direct control onto motor behaviors that comprise a feeding response^{28,57}. Regarding NAc-LH projections that mostly originate from the shell of the NAc, GABA signalling – notably originating from NAc D1-expressing neurons – has been shown to be increased during food intake, consequently terminating feeding, notably through decrease of orexin production⁵⁸. These findings also highlight that, beyond the core and shell subdivisions, subgroups of NAc D1-neurons likely control discrete behavioral dimensions, related to their location in subterritories of the NAc as well as their input and output projections. Finally, a main output structure of D1-expressing neurons of the NAc is the ventral pallidum (VP). Lesion of the VP – similarly to lesions of the LH - has been shown to counteract the orexigenic effect of NAc inhibition⁵⁹, suggesting a “pro-feeding” role of the VP. Accordingly, arypallidal neurons of the VP potentiate reward consumption⁶⁰. Even though, to the best of our knowledge, a direct implication of D1-neuron-to-VP projections in the regulation of food intake remains to be demonstrated, all these findings support that D1-neurons of the NAc can decrease food intake through several output pathways involving the LH, the VTA and the VP. Of note, even though we cannot exclude some effects related to spread of viral infections to the shell of the NAc, our chemogenetic manipulations were mostly restricted to the core subdivision, that sends projections to the VP and VTA but not the LH. The circuits by which D2-neurons of the NAc regulate feeding remain unclear. As mentioned, we cannot rule out that some of the effects we observed in this study might be related to an alteration in the activity of cholinergic interneurons (CIN) due the use of the D2-cre line - even though likely targeting at most a subset of ChAT-positive interneurons²². However, increasing CINs activity reduces palatable food consumption while their inhibition had the opposite effect⁶¹, in contrary to our findings with manipulations of D2-neurons. This suggests that most of the effects we observed were due to alterations of D2-MSNs. The sole output of this neuronal subpopulation is the VP. This raises the intriguing ideas that, if the VP is involved in the effects on feeding that we observed under manipulation of NAc subpopulations, it is likely that D1- and D2-neurons should project onto distinct VP neuronal subpopulations, though, to the best of our knowledge, this is an important aspect that remains to be demonstrated. Finally, the lateral inhibition between NAc subpopulations has been largely overlooked and is likely to participate to their differential roles, including feeding behavior.”

26- From the available data it is clear that the NAc neurons are important for feeding and energy balance, and that the same neuronal subtype of different subregions may play different roles. This could be better highlighted in the discussion.

We believe this aspect is addressed in the new discussion as shown in the preceding point (#25). This concept could be expanded though we could not due to space limitation.

27- Considering your working model, maybe it would be interesting to discuss these findings in the light of the known connections with the VTA and LH (or other appetite/energy balance brain regions)?

We believe we addressed this point above (#25) and now discuss this aspect in the revised version of the manuscript.

Minor comments

- I suggest separating the description of the different types of data in the first paragraph of results to make the reading easier.

We changed the description accordingly.

- I think that a summary table or a schematics representation of the findings would be very interesting to have, even if in sup. material.

We agree with the Reviewer and propose to provide a graphical abstract if the paper is accepted for publication.

- Legend of colors in graphs of Fig2A and E (it only appears in figures below)

The Figure has been changed (now Figure 2B) and the groups clearly indicated.

- Throughout the text, authors should refer to the supplementary figures with more detail: eg. Line 412 - Sup Fig1J, instead of SF1.

We apologize for not doing so in the initial version of the manuscript. This is now the case.

- Try to normalize axis whenever possible to allow better comparison between D1 and D2 effects: eg. Lever presses, breakpoint etc

We normalized axes whenever possible for better comparison.

- Line 223: through?

Corrected.

- Line 406: change amotivational

Corrected.

- Typo in S2D: salin instead of saline; refer to more details about this in methods.

Corrected.

- Authors should refer if the researchers were blind or not to the procedures.

This has now been added in the methods section (Line 528-530).

- Line 452 rephrase

This part of the manuscript has been largely re-written.

- Fig. 4D : using the nomenclature "after CNO" it is not clear as it looks like authors are referring to the period when CNO was removed. I would just put CNO period

We changed the nomenclature according to the Reviewer's suggestion.

- Overall, I suggest to improve figures representation (this is a personal comment, not a scientific one); especially photometry data.

We hope the Reviewer will agree the figures' representation has now been improved.

- Scale bars missing in figures (eg 3D, 5A...); 3D needs to be larger, the figure of the box is not very clear, maybe a legend would help

Scale bars are now added and Figure 3 has been changed.

- Fig4a missing time

This has been added.

- Include references of food, palatable pellets

The references are added.

- Line 492: refer that these measures were obtained by MRI.

This has been added.

- Maybe change Withdrawal with after CNO suspension or CNO removal?

This has been done.

- For all graphs include the individual dispersion points to highlight variability.

We added dispersion point for histograms, however it was not possible for graphs with repeated measures that would make the figure hard to read. We hope this will satisfy the Reviewer's concern.

- Provide better figures for 5A; mice scheme can be improved by adding the virus injected; figures missing scale bars and small, low quality, the whole NAc should be presented
Changes were done accordingly to the Reviewer's recommendations.

- Line 547-548: I don't think there are enough experiments to support this affirmation as it stands.

We now expanded this aspect regarding the potential (lack of) implication of leptin and ghrelin in our effects, notably in regard with the new data on leptin and ghrelin dosages (line 328-335).

Reviewer #2: Walle et al. are investigating how D1 and D2 accumbal core MSNs regulate behavior in an operant task. This is an important area of study as the precise push-pull of these cell populations in driving motivated behavior is not fully understood. While the authors show some exciting data suggesting that D1 and D2 MSNs in the NAc drive operant performance for food rewards, along with bidirectional modulation of feeding and wheel running, the study suffers from lack of hypothesis-driven rationale, overinterpretation of experimental results, and at times is missing control groups from experiments. Throughout the paper, experiments are presented and results stated, but often in the absence of specific hypotheses to inform the experimental plan, or without conclusions formed about the experimental results.

We thank the Reviewer for highlighting the importance of the area of study and considering that the data are exciting. We took the Reviewer's remarks, seriously and re-wrote the manuscript to better present the rationale for the experiments and not to overinterpret the results. The results section in particular has been rewritten such that we now have a hypothesis-driven rationale throughout the text which better reflects the logical order of experiments. In addition we performed additional experiments that make the set of data more coherent.

Additionally, while obesity and anorexia inarguably affect similar mesolimbic circuits, the premise of the paper, that obesity and AN are opposite ends of the spectrum is reductive.

We understand the Reviewer's concern. Only in some aspects, obesity and AN are opposite (BMI), but there are unique aspects in other domains where the comparison is irrelevant (body image disturbance). We based this statement on the work of other authors proposing that obesity resulting from compulsive overeating and anorexia nervosa are conditions at opposite ends of a continuum of extreme body weight disturbance that involve higher-order executive functions (see Foldi et al., 2021 PMID 34217755; Wierenga et al., 2014 PMID 25538579). In particular, evidence supports reduced behavioral control in obesity but elevated control of behavior in anorexia with respect to food and feeding. Nonetheless we understand the Reviewer's comment and toned-down this statement in the revised version of the manuscript (abstract, introduction and discussion).

For these reasons it is this reviewer's opinion that at this time this paper is not suitable for publication.

We hope the Reviewer will agree that we addressed her/his concerns and will now find the paper suitable for publication.

Concerns are outlined below:

1- Vague language is used in the abstract – for instance in lines 49-51 “effects” are referenced. We re-wrote the abstract to make it more descriptive of the results, without being extensive due to word limitation.

2- Line 106 – ref 34, they say chronic disruption of d1s leads to obesity. The reference in question showed that chronic inhibition of d1s is somewhat protective from DIO
We apologize for this mistake that has been corrected.

- For Figure 1, It looks like for the D1s, CNO might be the effect? Data from experiments using control GFP viruses should be presented and analyzed alongside the dreadd groups
The reason why we did not include a non-DREADD control group is due to the difficulty of generating enough transgenic animals at once to compare 3 groups (DREADDGi, DREADDGq, Reporter control) in parallel. We therefore decided to favor the comparison of DREADD Gi and Gq groups with within-subject comparisons (saline vs. CNO). Nonetheless, we indeed initially thought the decrease in performance could be due to a CNO effect since performance decreased in both Gi and Gq groups, though the fact that we found the opposite effect in the D2-Gi group goes against this hypothesis. However, to ensure this is not the case we performed an additional experiments: we used the recently-developed DREADD ligand JHU37160 (J60) in a cohort of D1-cre mice expressing either Gi or Gq DREADD and found the same effects – even though a bit milder – as for CNO. These data are now presented in supplementary figure 1H. Moreover, for the reviewers discretion, we recently performed an experiment in which inhibitory DREADDs were expressed in a subpopulation of dopaminergic neurons, with a control group expressing mCherry. Using CNO, we found an inhibitory effect on performance in the progressive ratio task, but no effect in the mCherry control group. These data will be part of another paper currently under preparation. We hope the Reviewer will agree that these new findings ensure that the effect in the D1-cre groups is not related to CNO but rather to the effects of DREADD.

- For data presented in Figure 2, the authors looked at how inhibition and activation of NAc d1s and d2s affected ad lib chow intake, and saw that inhibiting d2s decreased chow consumption up to 48 hours, while activating d2s only enhanced chow consumption 2 days later. Why do they think it would last two days but not have an affect initially? For instance, are they activating lasting plasticity mechanisms that are in turn driving hunger? This data should be further explained or removed from the narrative.

We thank the Reviewer for raising this important point that was also raised by the other Reviewer (point #9). As we originally explained in the manuscript, we believe that, when in their housing cage, the orexigenic effect of chemogenetic activation of D2-expressing neurons is initially overshadowed by the strong decrease in motivation. In other words, the drive to retrieve the chow pellets (climb on the grid to collect the food) is blunted. It is only afterwards that the orexigenic effect takes over, leading to increased consumption and weight gain. This is supported by our findings showing immediate increase in consumption when using a sweet milk solution delivered through a dipper, which requires little effort (now Fig. 2B). We now tried to explain better this aspect and wrote: “*We interpret this latter finding as a rebound effect resulting from an initial overshadowing of the orexigenic effect of D2-neurons activation by the blunted motivational drive, decreasing the willingness to exert effort to retrieve the food pellets despite the increase in hunger drive. This is supported by the observation that in the case of easily-accessible liquid food (milk), food consumption immediately increases as animals mostly stay at the dipper to consume (Figure 2B and SF2A)*” (line 168-173).

- Is the same dreadd cohort of mice used for experiments shown in figures 1-4? If so, correlative analyses could be used to explore whether there are relationships between variables, for instance whether performance on operant assays correlates with food consumption, etc.
We apologize for not making that clearer, it is now clarified in the methods. Different groups were used for these analyses. Therefore, we unfortunately cannot run correlative analyses, which, we agree, could have been really interesting. The reason why we preferred to use different cohorts was to avoid confounding effects due to repetition of CNO injections, periods of food deprivation and aging of the mice.

- They recorded bulk calcium activity via fiber optics in the nac during feeding or wheel running. The analysis as presented in fig 3e is referred to as revealing that during consumption d2 activity “Remains significantly higher” than d1s (line 470). But only perievent histograms are shown, but not an averaged data set on which statistics were run.

We added several analyses now presented in Figure 3 that further consolidate the findings. We analyzed AUC, average z-score, peak z-score and transient rate. All these new analyses further confirm the differences between relative bulk activity between D1- and D2-neurons during feeding and wheel running.

- Fig 3e-g have red, blue, and black horizontal lines on the graphs but it is not explained what these are.

We apologize for not making this clearer. We now added the legend on the figures (see figure 3A and B).

- In the manuscript, pubmed IDs instead of citations

We have checked that the references were appropriately cited.

- Zhu 2016 (frontiers) already showed that chemogenetic modulation of d1 and d2 nac neurons bidirectionally drives wheel running

The Reviewer is right and we cited the Zhu et al. paper in the original version of the manuscript. This work was indeed one of the studies that encouraged us to further explore the relative roles of NAc subpopulations in the balance between food intake and activity-mediated energy expenditure. We now clearly indicate that our findings in Figure 4 of the revised manuscript mostly recapitulate Zhu et al.'s findings (line 372-373). We hope that the Reviewer will agree that our study goes beyond the findings of Zhu et al., notably by demonstrating that biasing the balance between D1- and D2-neurons can trigger pathological-like phenotype - in particular AN-like phenotypes - by altering the equilibrium between food intake and activity-mediated energy expenditure. Moreover, we provide evidence of neuronal pattern supporting the differential activities of D1- and D2-neurons during running vs. feeding.

- Weight graphs should be plotted over time. So should distance travelled; right now just “food consumption” and “wheel running variation” is plotted (right now they are just listed in a supplemental table).

We added these data in supplementary figures 4 and 5. As the Reviewer will see, the reason why we presented “variations” is because the variability across animals is quite high, overshadowing individual variations. Nonetheless we agree with the Reviewer that raw data can be informative for potential reproduction of the findings by author investigators.

- Chronic DREADD manipulation is not advised for long-term experiments as they lose efficacy via receptor desensitization with frequent dosing over weeks (Roth 2016, PMID: 26889809).

The point raised by the Reviewer is crucial and has been extensively discussed in Claes et al. (2022, PMID 35406674). However, while some studies proposed desensitization after repeated treatment – often with relatively high doses of CNO – many did not. As pointed by Claes et al. (2022), this could be due to overexpression of the DREADD receptors that provides receptor reserve, therefore preventing desensitization.

While we cannot completely rule out some desensitization mechanisms, the observation that the majority of our effects (running and feeding) disappeared as early as 24 hours after CNO cessation support a certain efficacy of DREADD activation throughout chronic CNO treatment.

REVIEWERS' COMMENTS

Reviewer #1 (Remarks to the Author):

The authors have performed a very good work and answered to all of my comments in detail. I am satisfied with the additional data provided, namely the dLight measurements, measurement of peripheral molecules related to satiety, and the evaluation of D1- and D2-neurons in two models related to ED. I think that these findings strengthen the core of the manuscript. I just have a few minor comments that do not hamper the acceptance of the manuscript.

Minor comments:

- Considering the differential role of NAc core and Shell, I think it would be more adequate to name which region is being mostly targeted whenever its appropriate, including the abstract
- lines 39-40; 48-49 – Revise sentences
- line 446: VP is not the only output of D2-MSNs
- revise new text in discussion, some sentences can be improved.

Reviewer #2 (Remarks to the Author):

Walle et al. have made substantial improvements to this manuscript. In particular, the perievent histograms in figure 2 demonstrate the unique temporal roles of calcium (as a proxy for neural activity) vs dopamine binding during a lick. The only issue that remains is for the authors to please provide the weight curves over time, similarly to how they plot wheel running variation or food consumption (for example, Figure 4C). A minor point - line 152 references the wrong figure.

Reviewers' Comments:

Reviewer #1: The authors have performed a very good work and answered to all of my comments in detail. I am satisfied with the additional data provided, namely the dLight measurements, measurement of peripheral molecules related to satiety, and the evaluation of D1- and D2-neurons in two models related to ED. I think that these findings strengthen the core of the manuscript. I just have a few minor comments that do not hamper the acceptance of the manuscript.

We are grateful to the Reviewer for her appreciation of our work and for the very constructive comments throughout the review process that allowed us to significantly improve the manuscript.

Minor comments:

- Considering the differential role of NAc core and Shell, I think it would be more adequate to name which region is being mostly targeted whenever its appropriate, including the abstract
This has been changed, including in the abstract.

- lines 39-40; 48-49 – Revise sentences
Sentences were revised.

- line 446: VP is not the only output of D2-MSNs
The reviewer is right. We now wrote “the VP is a main output of D2-MSNs”.

- revise new text in discussion, some sentences can be improved.
We revised the text and improved the sentences.

Reviewer #2: Walle et al. have made substantial improvements to this manuscript. In particular, the perievent histograms in figure 2 demonstrate the unique temporal roles of calcium (as a proxy for neural activity) vs dopamine binding during a lick. The only issue that remains is for the authors to please provide the weight curves over time, similarly to how they plot wheel running variation or food consumption (for example, Figure 4C).

We thank the Reviewer for her/his very positive comments regarding the new experiments we provided in the revised manuscript and her/his constructive comments throughout the review process that helped improving the manuscript.

We now provide weight curves over time in revised supplementary figures 4L and 5F.

A minor point - line 152 references the wrong figure.
We apologize for this mistake that is now corrected.